# Learning Jump-Diffusion Dynamics from Irregularly-Sampled Data

## Abstract

Accurately modeling time-continuous stochastic processes from irregular observations remains a significant challenge. In this paper, we leverage ideas from generative modeling of image data to push the boundary of time series generation. For this, we find new generators of SDEs and jump processes for conditional interpolation which match the marginal distributions of the time series of interest. Specifically, we can handle discontinuities of the underlying processes by parameterizing the jump kernel densities by scaled Gaussians that allow for closed form formulas and hence rapid evaluation of the corresponding Kullback-Leibler divergence in the loss. Unlike most other approaches, we explicitly account for both irregular and non-aligned sampling times in constructing the generators. We also clarify several theoretical aspects that lead to a more robust formulation of the model. We underline our theoretical results by numerical experiments involving combinations of jumps and SDE dynamics that illustrate the benefits of the proposed framework

## 1 Introduction

Generative modeling has shown impressive results in imaging and natural sciences Corso et al. (2023); Rombach et al. (2022); Sohl-Dickstein et al. (2015); Song et al. (2021). However, the generation of time series, which arise in the context of modeling financial, meteorological and patient vital data has received less attention Bühler et al. (2020); Jia & Benson (2019); Kerrigan et al. (2024); Kidger et al. (2021a); Zhang et al. (2024). In particular, in the study of financial markets and price prediction Gülmez (2023); Wu et al. (2023), the (conditional) generation of time series data is an important cornerstone. For instance, in deep hedging Bühler et al. (2018), simulation of market data is crucial to learn appropriate hedging strategies. Often for price predictions, not only the most likely next price is of interest, but also its uncertainty. Therefore, in a Bayesian way we would like to model the prior of such time series data using neural networks.

The usual approaches to model the time series distribution involve adversarial training Kidger et al. (2021a), backpropagation through the SDE solver Jia & Benson (2019), or fitting an object as high dimensional as the number of discrete time points Kerrigan et al. (2024). Another difficulty that many methods face is, that they rely on fixed time discretizations, and therefore are not feasible e.g. for many medical datasets, where points are measured irregularly. Although it is sometimes possible to fill those missing values as in Yoon et al. (2019), we are interested in exploring a method which is not reliant on a fixed time grid and can directly accommodate irregular and varying observation times.

*Simulation-free* approaches which do not have to solve an ODE/SDE during training, such as flow matching Albergo et al. (2023); Lipman et al. (2023); Liu et al. (2023b); Wald & Steidl (2025) or score-based diffusion Song et al. (2021) have significantly improved the performance of generative models compared to adversarial networks Goodfellow et al. (2014) or simulation-based methods Chen et al. (2018). Recently, such simulation-free methods have been applied to time series generation with regular observation times employing a high-dimensional latent distribution in Kerrigan et al. (2024). In contrast, the authors of Zhang et al. (2024) proposed to view time series generation as essentially generating one-dimensional curves via interpolating between the discretization points. This interpolation is done by learning an SDE between the discrete time points and taking the realizations at previous time points as conditions. A very similar idea has been proposed by Chen et al.

(2024) in the language of stochastic interpolants and applied to more challenging high-dimensional datasets, and a similar algorithm is proposed in El-Gazzar & van Gerven (2025). During inference, the autoregressive generation is akin to doing the next word prediction in GPT models Vaswani et al. (2017) highlighting the conceptual similarity between sequential text generation and sequential time series synthesis, while differing crucially in that the latter must handle irregular temporal spacing

However, processes in economics are often fundamentally discontinuous as buying or selling a stock may induce a jump. Jump processes have been already employed in Campbell et al. (2023) for the forward process inside a diffusion framework Song et al. (2021), in Benton et al. (2024) for parameterizing and learning generators on discrete state spaces via Markov chains or learning Schrödinger bridges via h-transforms, see Zlotchevski & Chen (2025). Based on the fact that the generator of a Markov process can be exhaustively decomposed into an SDE and a jump part, see von Waldenfels (1965), Holderrieth et al. (2025) considered generator matching for static and structured problems, e.g. image and protein generation.

In this paper, we combine generator matching with the dynamics of time series generation which poses both theoretical and numerical challenges. Since the approach of Zhang et al. (2024) has a singularity when reaching the final time point, which is detrimental in either training or inference, we modify it by allowing small amounts of Gaussian noise at known discretization points, stabilizing the Brownian bridges. For this specific Markov process, we calculate the generator of the now stabilized SDE and find an expression for the jump part of the generator, allowing us to combine the SDE and jump part via the Markov superposition principle. In contrast to the paper of Holderrieth et al. (2025), we parameterize the jumps not via binning, but by Gaussian-like functions. This allows to give analytical expression for the Kullback-Leibler divergence used in the corresponding loss function. Since we have to handle non Markov time series, we need to include memory as conditions in our next point predictors. Here we prove that our learned time series approximates the true joint distribution and does not only capture marginals. In the numerical section, we verify our method on several data sets and compare it with state of the art time series generation methods. Moreover, we show that our approach is also working for high-dimensional data, see Section 6.2.

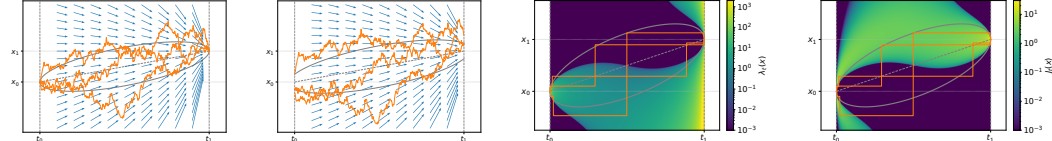

Figure 1: On the left side we see three paths of the interpolating diffusion bridge between $\mathcal{N}(x_0, \rho^2)$ and $\mathcal{N}(x_1, \rho^2)$, for small (leftmost) and large $\rho$ (left) with the underlying conditional drift (blue), it is evident that the latter drift field is less singular at the end point. On the right side we see two paths of the interpolating jump bridge together with heat maps of the conditional jump intensity $\lambda_t$ (right) and target distribution $J_t$ (rightmost). In all cases we depict sample paths (orange) and marginal distribution (mean and standard deviation indicated by gray lines)

**Contributions.** Our work provides a new perspective for understanding and generating autoregressive time series. For the first time, we construct a fully flexible generative model for sampling time series given their samples at possibly different and irregular times through extending and refining generator matching techniques such that they become accessible for conditional trajectory interpolation. Here are our main contributions:

- Additionally to diffusion bridges considered by Zhang et al. (2024), we introduce jump processes for interpolation and derive stabilized paths in Proposition 2, avoiding singularities for both classes, see Figure 1 for an illustration. This is key for making training stable and it increases the methods flexibility to account for noncontinuous trajectories.

- We propose to parametrize the density of the jump kernel by scaled Gaussians and to learn the corresponding parameters. This allows us to derive closed-form expressions for the Kullback-Leibler loss function, as presented in Proposition 5. These expressions can be computed efficiently via Proposition 6, avoiding the expensive binning method used in Holderrieth et al. (2025) and makes training scalable.

- We rigorously derive all our results for irregular time series, and offer a first rigorous perspective why the memory mechanism works and is robust, see Proposition 7.

Although we also design a practical algorithm, which we validate through numerical experiments in various dimensions, our paper mainly focuses on rigorous understanding and derivations of the mathematical objects to set up a basis for future work. For readers seeking an intuitive summary without mathematical nuances, Appendix B provides a high-level overview of the method. Additional definitions are collected in Appendix C, all proofs are presented in Appendix D, additional results with real world data and implementation details are given in in Appendices F and G.

## 2 PROBLEM SETTING

We are given $N$ time series $\mathcal{X} := \{x^1, ..., x^N\}$, $x^i := (x^i_{t^i_j})^{n_i}_{j=0} \in \mathbb{R}^d$ sampled at different times $0 = t^i_0 < t^i_1 < \ldots < t^i_{n_i} = T$ which are i.i.d observations at discrete times of realizations of a random process $X = (X_t)_{t \in [0,T]}$. Such data might be encountered in high frequency financial markets, since trades, placing and executing of orders can occur at any point in time, or in the context of clinical trajectories of patients' vital signs evolution.

We assume that $X$ is a random variable in the function space of all right continuous functions in $t$ with left limits. Then, the distribution of $X$ is uniquely determined by all finite-dimensional marginals (Billingsley, 2013, Section 13), so that we aim to learn the latter ones. More precisely, for given times $0 = t_0 < t_1 < ... < t_n = T$ (not necessarily the above ones), we want to sample from $P_{X_{t_0}, ..., X_{t_n}}$ by learning neural networks based on our data in $\mathcal{X}$. Using disintegration, see Appendix C, we know that

$$P_{X_{t_0}, ..., X_{t_n}} = P_{X_{t_n} | X_{t_{n-1}} = x_{n-1}, ..., X_{t_0} = x_0} \times_{x_{n-1}, ..., x_0} \cdots \times_{x_1, x_0} P_{X_{t_1} | X_{t_0} = x_0} \times_{x_0} P_{X_{t_0}}.$$

Hence, in order to generate a new time series, we start with an initial value $x_0 \sim P_{X_{t_0}}$ and then draw inductively $x_{j+1} \sim P_{X_{t_{j+1}} | X_{t_j}, ..., X_{t_0}}$. In order to sample from the latter distribution, we will learn for $x_0, \ldots, x_j \in \mathbb{R}^d$ a Markov process $(Y^{x_0, ..., x_j}_t)_{t \in [t_j, t_{j+1}]}$ starting in $Y^{x_0, ..., x_j}_{t_j} = x_j$ such that approximately

$$Y^{x_0, ..., x_j}_{t_{j+1}} \sim P_{X_{t_{j+1}} | X_{t_j} = x_j, ..., X_{t_0} = x_0}. \tag{1}$$

For learning this interpolating Markov process we employ ideas from generator matching which we explain in the next section.

## 3 MARKOV PROCESSES VIA GENERATORS

After a brief recall of Markov processes and their generators, we propose simple Markov processes with drift-diffusion and jump generators that can be found analytically in Proposition 2i) and ii), respectively. Note that in contrast to the Markov process with drift-diffusion generator in Proposition 2iii) suggested in Zhang et al. (2024), we do not have to cope with singularity issues and our process in Proposition 2i) is a stabilized version of the Brownian bridge. We first introduce Markov proceesses, generators and the corresponding Kolmogorov forward equation, based on Holderrieth et al. (2025).

Let $\mathcal{B}(\mathbb{R}^d)$ denote the Borel sets on $\mathbb{R}^d$ and $\mathcal{B}_b(\mathbb{R}^d)$ the space of bounded, (Borel-)measurable functions on $\mathbb{R}^d$. Further, let $\mathcal{M}_+(\mathbb{R}^d)$ denote the space of non-negative measures on $\mathbb{R}^d$, $\mathcal{P}(\mathbb{R}^d)$ the probability measures and $\mathcal{P}_2(\mathbb{R}^d)$ the probability measures with finite second moments.

We consider a Markov process $(Y_t)_{t \in [0,1]}$ in $\mathbb{R}^d$. The process can be described by its transition semigroup, which acts on test functions and is the continuous analogue of transition matrices of discrete Markov chains (in the sense that together with the initial distribution at $t = 0$, it completely determines the distribution of the process). The transition semigroup and hence the process can be characterized by its generator $(\mathcal{L}_t)_t$, mapping from a subset of $\mathcal{B}_b(\mathbb{R}^d)$ into the measurable functions by

$$\mathcal{L}_t f(x) := \lim_{h \downarrow 0} \frac{\mathbb{E}[f(Y_{t+h}) - f(Y_t) | Y_t = x]}{h}.$$

It can be rewritten in the form Holderrieth et al. (2025); von Waldenfels (1965)

$$\mathcal{L}_t f(x) = \underbrace{\langle \nabla_x f(x), u_t(x) \rangle}_{\text{drift}} + \underbrace{\frac{1}{2} \langle \Sigma_t^2(x), \Delta f(x) \rangle}_{\text{diffusion}} + \underbrace{\int (f(y) - f(x)) \, Q_t(\mathrm{d}y, x)}_{\text{jump}},$$

where $u_t : \mathbb{R}^d \to \mathbb{R}^d$ is the drift, $\Sigma_t : \mathbb{R}^d \to \mathbb{R}^{2d}$ is the diffusion coefficient and $Q_t : \mathcal{B}(\mathbb{R}^d) \times \mathbb{R}^d \to \mathbb{R}$ is the rate (jump) kernel (i.e., $x \mapsto Q_t(B, x)$ is measurable and $B \mapsto Q_t(B, x)$ is a finite positive measure). If $X_t$ has law $P_t$ with density $p_t$ and $Q_t(\cdot, x) \in \mathcal{M}_+(\mathbb{R}^d)$ are given by Lebesgue densities $q_t(y, x) \, \mathrm{d}y$, then the *Kolmogorov forward equation* (KFE)

$$\partial_t \langle p_t, f \rangle = \langle p_t, \mathcal{L}_t f \rangle$$

holds true. In this paper, we restrict our attention to $\Sigma_t^2 = \eta^2 I_d$ with an appropriately given $\eta > 0$. Then, taking the adjoint of the KFE provides the following PDE

$$\partial_t p_t = -\nabla \cdot (p_t u_t) + \frac{1}{2} \eta^2 \Delta p_t + \int p_t(y) q_t(x, y) - p_t(x) q_t(y, x) \, \mathrm{d}y.$$

Given a generator $\mathcal{L}_t$ which we address in the following by the triple $(u_t, \eta, q_t)$, we can approximate a corresponding Markov process by the following Algorithm 1, see (Holderrieth et al., 2025, Appendix B).

---

**Algorithm 1** Approximating a Markov process with generator $\mathcal{L}_t = (u_t, \eta, q_t)$

---

**Given:** $u_t, \eta, q_t(\cdot, x) = \lambda_t(x) J_t(\cdot, x), \lambda_t \geq 0, J_t(\cdot, x) \in \mathcal{P}(\mathbb{R}^d)$
**Initialization:** $x_0 \sim X_0$, step size $h := \frac{1}{n}$

0: **for** $t$ in $\{\frac{0}{n}, \frac{1}{n}, \ldots, \frac{n-1}{n}\}$ **do**
0:     $x_{t+h}^J \sim J_t(\cdot, x_t)$
0:     $m \sim \text{Bernoulli}(h\lambda_t(x_t)), \epsilon_t \sim \mathcal{N}(0, 1)$
0:     $x_{t+h}^D = x_t + h u_t(x_t) + \sqrt{h} \, \eta \, (x_t) \epsilon_t$
0:     $x_{t+h} = m x_{t+h}^J + (1-m) x_{t+h}^D$
0: **end for**=0
**Return:** $x_1$ approximately distributed as $X_1$

---

**Remark 1.** *Note that for $u_t = 0, \sigma_t = 0$ we have $x_{t+h}^D = x_t$ and thus obtain a pure jump process. On the other hand, for $\lambda_t = 0$ we have that $m = 0$ and thus $x_{t+h} = x_{t+h}^D$, a pure diffusion process.*

Our learning of the conditional distribution is based on the following time-inhomogenous Markov processes, whose generators we can give explicitly.

**Proposition 2.** *For arbitrary fixed $x_0, x_1 \in \mathbb{R}^d$, set $m_t := (1-t)x_0 + tx_1$ and $\tau_t := \eta^2 t(1-t) + \rho^2$ with $\eta > 0$. Let $P_t = P_t(\cdot, x_0, x_1) = \mathcal{N}(m_t, \tau_t)$ with density $p_t$. Then each of the following generators $\mathcal{L}_t^{\text{Diff}} = \mathcal{L}_t^{\text{Diff}, x_0, x_1}$ and $\mathcal{L}_t^{\text{Jump}} = \mathcal{L}_t^{\text{Jump}, x_0, x_1}$ fulfills the KFE and thus give rise to a Markov process $Y_t = Y_t^{x_0, x_1}$ with marginals $Y_t \sim P_t$:*

> i) $\mathcal{L}_t^{\text{Diff}} f(x) = \nabla f(x)^{\mathsf{T}} u_t(x) + \frac{\eta^2}{2} \Delta f(x)$, where $u_t^{x_0, x_1} = u_t := x_1 - x_0 - (x - m_t) \frac{\eta^2 t}{\tau_t}$.
> *An associated Markov process is given by the solution of the SDE*
>
> $$\mathrm{d}Y_t = u_t \, \mathrm{d}t + \eta \, \mathrm{d}W_t; \quad Y_0 = x_0 \sim \mathcal{N}(x_0, \rho^2).$$

> ii) $\mathcal{L}_t^{\text{Jump}} f(x) = \int \big( f(y) - f(x) \big) q_t(y, x) \, \mathrm{d}y$, where $q_t^{x_0, x_1}(y, x) = q_t(y, x) =:= \lambda_t(x) J_t(y)$, *and*
>
> $$J_t := \frac{\max(0, \xi_t) p_t}{\int \max(0, \xi_t) p_t \, \mathrm{d}y}, \quad \lambda_t := \max(0, -\xi_t),$$
>
> $$\xi_t(y) := \frac{\eta^2 (1 - 2t)}{2\tau_t} \Big( \frac{\|y - m_t\|^2}{\tau_t} + 2 a_t^{\mathsf{T}} \frac{y - m_t}{\sqrt{\tau_t}} - d \Big), \quad a_t := \frac{\sqrt{\tau_t}}{\eta^2 (1 - 2t)} (x_1 - x_0).$$

*iii) If $\rho = 0$, then $\mathcal{L}_t^{\text{Diff}}$ in i) simplifies towards $u_t(x) = \frac{x_1 - x}{1-t}$ and the SDE becomes*

$$dY_t = \frac{x_1 - Y_t}{1 - t}\, dt + \eta\, dW_t, \quad Y_0 = x_0.$$

For $\rho = 0$, we recover the drift of Zhang et al. (2024) which is singular at $t = 1$. Moreover, $P_0(\cdot, x_0, x_1) = \delta_{x_0}, P_0(\cdot, x_0, x_1) = \delta_{x_1}$, and we see that the Markov process generated by $\mathcal{L}_t^{\text{Diff}}$ is a Brownian bridge connecting $x_0$ and $x_1$. For $\rho > 0$, we don't encounter singularities and have $P_0(\cdot, x_0, x_1) = \mathcal{N}(x_0, \rho^2), \quad P_1(\cdot, x_0, x_1) = \mathcal{N}(x_1, \rho^2)$. The interpolating diffusion and jump bridges are illustrated in Figure 1.

## 4 LEARNING THE DRIFT AND THE JUMP KERNEL

In this section, we employ the simple Markov processes from the previous section to train a neural network which generates samples from the conditional distributions (1).

We start by describing the procedure for the first conditional distribution, $P_{X_{t_1}|X_{t_0}=x_0}$, where $x_0 \in \mathbb{R}^d$ is arbitrarily fixed, and assume for simplicity that $t_0 = 0$ and $t_1 = 1$. The strategy is to construct Markov processes $(Y_t^{x_0})_{t \in [0,1]}$ with $Y_0^{x_0} = x_0$ and $Y_1^{x_0} \sim P_{X_1|X_0=x_0}$ by learning their generators. Then we can sample from $P_{X_1|X_0=x_0}$ using Algorithm 1.

We start with a family $(P_t)_{t \in [0,1]}$ of Markov kernels $P_t(\cdot, \cdot, \cdot) : \mathcal{B}(\mathbb{R}^d) \times \mathbb{R}^d \times \mathbb{R}^d \to [0, 1]$ with

$$P_0(\cdot, x_0, x_1) = \delta_{x_0} \quad \text{and} \quad P_1(\cdot, x_0, x_1) = \delta_{x_1}. \tag{2}$$

For example, the kernels $P_t(\cdot, x_0, x_1) = \mathcal{N}(m_t, \tau_t)$ in Proposition 2 have this property for $\rho = 0$, while they fulfill it "nearly" for small $\rho > 0$, see Remark 3 below for the appropriate statement in this case. We consider

$$\alpha_t^{x_0} \coloneqq P_t(\,dx, x_0, x_1) \times_{x_1} P_{X_1|X_0=x_0}(\,dx_1) \in \mathcal{P}_2(\mathbb{R}^d \times \mathbb{R}^d) \quad \text{and} \quad P_t^{x_0} \coloneqq \pi_\sharp^x(\alpha_t^{x_0}).$$

Clearly, we have by definition that $P_0^{x_0} = \delta_{x_0}$ and $P_1^{x_0} = P_{X_1|X_0=x_0}$. Thus, it remains to find a process $Y_t^{x_0} \sim P_t^{x_0}$ via its generator $\mathcal{L}_t^{x_0}$. Let $\mathcal{L}_t^{x_0,x_1}$ be generators of processes $Y_t^{x_0,x_1}$ with $Y_t^{x_0,x_1} \sim P_t(\cdot, x_0, x_1)$. Then it is shown in (Holderrieth et al., 2025, Proposition 1), that $\mathcal{L}_t^{x_0}$ defined by

$$\mathcal{L}_t^{x_0}(f)(x) \coloneqq \int \mathcal{L}_t^{x_0,x_1}(f)(x)\, d(\alpha_t^{x_0})_x(x_1) \tag{3}$$

generates a Markov process $Y_t^{x_0}$ with $Y_t^{x_0} \sim P_t^{x_0}$.

**Remark 3.** *Instead of* (2)*, we will start with Markov kernels $P_t(\cdot, x_0, x_1)$ with*

$$P_0(\cdot, x_0, x_1) = \mathcal{N}(x_0, \rho^2) \quad \text{and} \quad P_1(\cdot, x_0, x_1) = \mathcal{N}(x_1, \rho^2), \quad \rho > 0$$

*Let $\mathcal{L}_t^{x_0}$ be the generator from* (3) *and let $Y_t^{x_0}$ be an associated Markov process. Then we have $Y_0^{x_0} \sim \mathcal{N}(x_0, \rho^2)$ and $Y_1^{x_0} \sim P_{X_1|X_0=x_0} * \mathcal{N}(0, \rho^2)$, see Appendix D.*

We want to approximate $\mathcal{L}_t^{x_0}$ by a neural network based on the known analytical expressions of the generator $\mathcal{L}_t^{x_0,x_1}$ from Proposition 2. More precisely, we will learn two neural networks, one for approximating a drift-diffusion process via a learned drift $v_t^{x_0,\theta}$ and fixed diffusion, and one for approximating a jump process via a learned jump kernel $r_t^{x_0,\theta}$. Finally, we can take a convex combination of both processes as in Holderrieth et al. (2025). Next, we have the following observation from (Holderrieth et al., 2025, Proposition 2). This shows that learning the conditional generator loss indeed yields the velocity and jump kernel minimizing certain Bregman divergences. Further, any convex combinations of the resulting generators is a viable generator, which is dubbed "Markov superposition principle" in Holderrieth et al. (2025).

**Proposition 4.** *Let $x_0 \in \mathbb{R}^d$ be arbitrarily fixed. Let $\mathcal{L}_t^{x_0}$ defined in* (3) *be either a drift-diffusion process with drift $u_t^{x_0}$ and diffusion $\eta$ or a jump process with rate kernel $q_t^{x_0}$. Let $u_t^{x_0,x_1}$ and $q_t^{x_0,x_1}$ be the corresponding terms of $\mathcal{L}_t^{x_0,x_1}$. Then it holds*

$$u_t^{x_0} = \underset{v_t^{x_0} \in L_2(P_t^{x_0})}{\arg\min}\; \mathbb{E}_{t \in [0,1], x_1 \sim P_{X_1|X_0=x_0}, x \sim P_t(\cdot, x_0, x_1)} \left[ \| u_t^{x_0,x_1}(x) - v_t^{x_0}(x) \|^2 \right]. \tag{4}$$

*and*

$$q_t^{x_0} = \arg\min_{r_t^{x_0}} \mathbb{E}_{t\in[0,1],x_1\sim P_{X_1|X_0=x_0},x\sim P_t(\cdot,x_0,x_1)}\left[D_{\mathrm{KL}}\left(q_t^{x_0,x_1}(\cdot,x),r_t^{x_0}(\cdot,x)\right)\right]. \tag{5}$$

*with the Kullback-Leibler divergence $D_{\mathrm{KL}}$ defined in Appendix C. Both $u_t^{x_0}$ and $q_t^{x_0}$ as well as any convex combination of the corresponding generators*

$$\mathcal{L}_{t,\alpha}^{x_0} f(x) := \alpha\left(\langle\nabla f(x), u_t^{x_0}(x)\rangle + \frac{1}{2}\eta^2\Delta f(x)\right) + (1-\alpha)\int(f(y)-f(x))q_t^{x_0}(\,\mathrm{d}y,x)$$

*for $\alpha \in [0,1]$ have the marginal distribution $P_t^{x_0}$.*

In Holderrieth et al. (2025), the authors do not intend to sample from a conditional distribution or time series data. Instead, their goal is to sample directly from a target distribution $P_{X_1}$, rather than from an intractable conditional distribution $P_{X_1|X_0=x_0}$. Hence, they can just train neural networks $v_t^{x_0,\theta}$ and $r_t^{x_0,\theta}$ with the loss functions (4) and (5).

In view of Proposition 4, for the drift-diffusion interpolation allowing us to sample from $P_{X_1|X_0}$, we suggest to minimize

$$\mathcal{E}^{\mathrm{Diff}}(\theta) := \mathbb{E}_{t\sim[0,1],(x_0,x_1)\sim P_{X_0,X_1},x\sim P_t(\cdot,x_0,x_1)}\left[\|u_t^{x_0,x_1}(x) - v_t^{x_0,\theta}(x)\|^2\right]$$

$$= \mathbb{E}_{t\sim[0,1],(x,x_0,x_1)\sim P_t(\cdot,x_0,x_1)\times_{x_0,x_1} P_{X_0,X_1}}\left[\|u_t^{x_0,x_1}(x) - v_t^{x_0,\theta}(x)\|^2\right].$$

This requires for inserting a condition into the neural network, which summarizes the path, i.e., some kind of memory.

Learning the jump interpolation is more involved since $r_t^{x_0,\theta}(\cdot,x)$ is a function and not just a vector as in the drift-diffusion case. We start by considering

$$\mathbb{E}_{t\sim[0,1],(x,x_0,x_1)\sim P_t(\cdot,x_0,x_1)\times_{x_0,x_1} P_{X_0,X_1}}\left[D_{\mathrm{KL}}\left(q_t^{x_0,x_1}(\cdot,x),r_t^{x_0,\theta}(\cdot,x)\right)\right]$$

The authors in (Holderrieth et al., 2025, Appendix F) suggest to approximate the function $r_t^{x_0,\theta}$ by dividing $\mathbb{R}$ into $n_b$ bins which results in an output dimension of $n_b\, d$ on $\mathbb{R}^d$ and is computationally demanding, since they learn each pixel independently. In this paper, we circumvent this difficulty by restricting our attention to functions $r_t^{x_0,\theta}$ of the form $\lambda\mathcal{N}(\mu,\Sigma)$. Interestingly, by the following proposition, the KL-divergence between $q_t^{x_0,x_1}(\cdot,x)$ and such functions can be directly computed.

**Proposition 5.** *Let $q_t^{x_0,x_1}(\cdot,x) = \lambda_t(x)J_t(\cdot)$ be defined by Proposition 2ii) with $\rho > 0$, and denote by $\mu^J$ and $\Sigma^J$ the mean and covariance matrix of $J_t$. Then it holds for fixed $t, x \in [0,1]\times\mathbb{R}^d$ that*

$$D_{\mathrm{KL}}(q_t^{x_0,x_1}(\cdot,x),\lambda\mathcal{N}(\mu,\Sigma)) = F_{t,x}(\lambda,\mu,\Sigma) + \mathrm{const},$$

$$F_{t,x}(\lambda,\mu,\Sigma) := \lambda + \lambda_t(x)\left(d\log(\det\Sigma) - \log(\lambda) + \frac{1}{2}\langle\Sigma^{-1},\Sigma^J\rangle + \frac{1}{2}\langle\mu^J - \mu,\Sigma^{-1}(\mu^J - \mu)\rangle\right),$$

*where $\mathrm{const}$ is a constant independent of $(\lambda,\mu,\Sigma)$. As a side result, for $\lambda_t(x) \neq 0$ the problem*

$$\min_{(\lambda,\mu,\sigma^2)\in\mathbb{R}_{>0}\times\mathbb{R}^d\times\mathbb{R}_{>0}} F_{t,x}(\lambda,\mu,\sigma^2 \mathrm{I}_d)$$

*has the unique minimizer $(\lambda_t(x),\mu^J,\mathrm{trace}\,\frac{\Sigma^J}{d})$.*

Our numerical considerations are done in dimensions $d = \{1,2\}$. In order to set up $F_{t,x}(\lambda,\mu,\Sigma)$, we need to compute the mean and covariance matrix $(\mu^J,\Sigma^J)$ of $J_t$. Surprisingly, we can derive closed-form formulas for these in the following proposition, which is related to but not contained in Bunne et al. (2023).

**Proposition 6.** *For dimensions $d \in \{1,2\}$ the mean $\mu^J = \mu_{x_0,x_1}^J$ and variance $\Sigma^J = \Sigma_{x_0,x_1}^J$ of $J_t$ can be computed in closed-form, see (8) and (11).*

Finally, we learn the jump process allowing us to sample from $P_{X_1|X_0=x_0}$ by minimizing the loss

$$\mathcal{E}^{\mathrm{Jump}}(\theta_1,\theta_2,\theta_3) := \mathbb{E}_{t\sim[0,1],(x,x_0,x_1)\sim P_t(\cdot,x_0,x_1)\times_{x_0,x_1} P_{X_0,X_1}}$$

$$\left[\lambda_t^{x_0,\theta_1}(x) + \lambda_t^{x_0,x_1}(x)\left(d\log(\sigma_t^{x_0,\theta_3}(x)) - \log(\lambda_t^{x_0,\theta_1}(x)) + \frac{(\sigma_t^{x_0,x_1,J})^2}{2\sigma_t^{x_0,\theta_3}(x)^2} + \frac{\|\mu_t^{x_0,x_1,J} - \mu_t^{x_0,\theta_2}(x)\|^2}{2\sigma_t^{x_0,\theta_3}(x)^2}\right)\right].$$

## 5 LEARNING IRREGULAR TIME SERIES

Now we show how to learn the generators of Markov processes using the given data $\mathcal{X}$ and then explain how to approximately sample from $X = (X_t)_t$. In particular, we approximate generators $\mathcal{L}_t^{((x_0,t_0),\ldots,(x_j,t_j),t_{j+1})}$ with associated Markov processes $Y_t^{((x_0,t_0),\ldots,(x_j,t_j),t_{j+1})}$ for $t \in [t_j, t_{j+1}]$ which fulfill $Y_{t_i}^{((x_0,t_0),\ldots,(x_j,t_j),t_{j+1})} \sim \mathcal{N}(x_j, \rho^2)$ and $Y_{t_{i+1}}^{((x_0,t_0),\ldots,(x_j,t_j),t_{j+1})} \sim P_{X_{t_{j+1}}|X_{t_0}=x_0,\ldots,X_{t_j}=x_j} * \mathcal{N}(0, \rho^2)$. To this end, we need the results from Proposition 2 for general time intervals $[t_j, t_{j+1}]$ given in its proof in Appendix D.

For a time series $x^i \in \mathcal{X}$ let $x_j^i := x_{t_j^i}^i$. We define the memory $\xi_j^i := \big((x_0^i, t_0^i), \ldots, (x_j^i, t_j^i)\big)$. Markov processes are analytically tractable, which is crucial for our construction, yet their expressiveness is limited. Augmenting them with a memory provides the flexibility needed to model systems whose future evolution depends on longer temporal context. To clarify the mechanism, the memory is simply the part of the past trajectory on which the model conditions when predicting the next step. Whereas a Markov model would only use the current state, our construction includes a controlled window of historic observations, which is essential for capturing non-Markovian dependencies, handling irregular sampling where time stamps themselves carry information, and modeling path-dependent behaviour. In practice, we restrict the memory to a window of a certain length, so that it represents a concise summary of the recent trajectory segment rather than the entire history.

For $t \in [0, T]$, there exists $j \in \{0, \ldots, n_i - 1\}$ such that $t \in [t_j^i, t_{j+1}^i)$. We denote this dependence by $j = j(t, i)$. Then we learn the network for the drift by minimizing the loss function

$$\mathcal{E}^{\text{Diff}}(\theta) := \mathbb{E}_{t \sim \mathcal{U}([0,T]), i \sim \mathcal{U}([N]), x \sim P_t(\cdot, x_j^i, x_{j+1}^i)} \Big[\big\| u_t^{x_j^i, x_{j+1}^i}(x) - v_t^{\xi_j^i, t_{j+1}^i, \theta}(x)\big\|^2\Big],$$

where $u_t^{x_j^i, x_{j+1}^i}$ is the generator of the Markov process connecting $x_{j(t,i)}^i$ and $x_{j(t,i)+1}^i$ from Proposition 2i). Similarly, we set up a network to learn the jump rate kernel with loss

$$\mathcal{E}^{\text{Jump}}(\theta) := \mathbb{E}_{t \sim \mathcal{U}([0,T]), i \sim \mathcal{U}([N]), x \sim P_t(\cdot, x_j^i, x_{j+1}^i)} \Big[\text{KL}\big(q_t^{x_j^i, x_{j+1}^i}(x), r_t^{\xi_j^i, t_{j+1}^i, \theta}(x)\big)\Big],$$

where $q_t^{x_j^i, x_{j+1}^i}$ is the generator of the Markov process connecting $x_j^i$ and $x_{j+1}^i$ from Proposition 2ii). Further, we restrict our network to parametrize Gaussian-like jump measures of the form $\lambda \mathcal{N}(\mu, \sigma^2 \, \text{Id}_d)$ such that we can compute the KL-divergence in closed form by Proposition 5. Once our generators $\mathcal{L}_t^{\cdot,\cdot} = \mathcal{L}_t^{\cdot,\cdot,\theta}$ of our Markov processes are learned, we sample our time series according to Algorithm 2, see Figure 2 right hand side. This is based on Algorithm 1 with time intervals $[t_i, t_{i+1}]$, $i = 0, \ldots, n-1$ instead of $[0, 1]$. In the supplemental material, we provide an illustration that shows how time series are generated successively using Algorithm 2.

Finally, the following result shows that the error of the finite-dimensional marginals can be bounded by the errors of the conditional transitions.

**Proposition 7** (informal, full statement in Appendix E.)**.** *Assume that the conditional law $P_{X_{t_{j+1}}|X_{t_0}=x_0,\ldots,X_{t_j}=x_j}$ is Lipschitz continuous in $(x_0, \ldots, x_j)$ with respect to the Wasserstein-2 metric. Then, if we can bound the approximation error for these conditional distributions for $j = 0, \ldots, n-1$, we also obtain a bound on the error of approximating the joint law $P_{X_{t_0},\ldots,X_{t_n}}$ in the Wasserstein-2-metric.*

## 6 NUMERICAL ILLUSTRATIONS

We first evaluate our approach on synthetic low-dimensional time series (non-Markovian trends and Black-Scholes stock data) under varying subsampling regimes, comparing against state-of-the-art baselines. For high-dimensional data, we introduce a moving-box video dataset, where SDE and jump components capture complementary aspects of the dynamics, and convex combinations balance their strengths. Following a reviewer's suggestion we moreover apply our approach to the ETTh1 dataset Zhou et al. (2021b), those results are reported in Appendix F.

---

**Algorithm 2** Sampling from glued Markov processes at $\tau = (t_0, \ldots, t_n)$.

---

**Given:** $\mathcal{L}_t^{\cdot,\cdot}$ and $x_0 \sim X_0$, $\tau = (t_0, \ldots, t_n)$. $\text{ALG}_1$ from Algorithm 1.

**Initialize:** $\xi_0 = (x_0, t_0)$

  0: **for** $i$ in $\{0, \ldots, n-1\}$ **do**

  0:     $x_{i+1} = \text{ALG}_1(x_i, \mathcal{L}_t^{\xi_i, t_{i+1}})$

  0:     $\xi_{i+1} = (\xi_i, (x_{i+1}, t_{i+1}))$

  0: **end for**=0

**Return:** $(x_0, \ldots, x_n)$ approximately distributed as $(X_{t_0}, \ldots, X_{t_n})$.

---

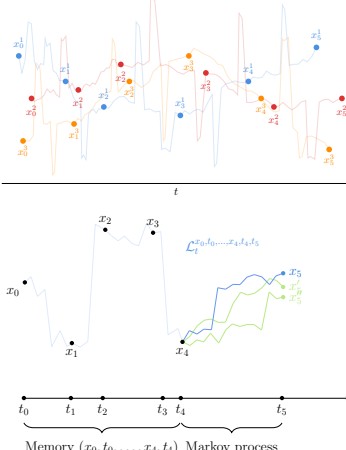

Figure 2: **Upper right:** Three example time series from a dataset. Only irregular observations (dots) are available; the underlying continuous-time paths between observations drawn in light colour for illustration are not observed. **Lower right:** Illustration of our generative model. A new trajectory is produced sequentially in time. For $t \in [t_4, t_5]$, the model receives the memory $(x_0, t_0), \ldots, (x_4, t_4)$ as well as the time point $t_5$. Two admissible continuations (green) are shown as different realizations of the local bridge; the sampled continuation is shown in blue.

## 6.1 LOW-DIMENSIONAL TIME SERIES

In our experiments we do not use the whole history. Instead we fix a memory length $m$ and consider $\mathcal{L}_t^{(x_{j-m}, t_{j-m}, \ldots, (x_j, t_j)), t_{j+1}, \theta}$ where we set $(x_k, t_k) = (x_0, 0)$ for $k < 0$. This is justified if the process $X_t$ we want to approximate mainly depends on time points $s < t$ close to $t$ and not on the whole interval $[0, t]$. Our code is written in PyTorch Paszke et al. (2019).

As a first synthetic data set, see Figure 3, we generate non-Markovian trajectories, which follow one of two deterministic trends, occasionally perturbed by small jumps, with 50 time steps. A detailed description is given in Section G.

As a second dataset we test our algorithms on synthetically generated stock data generated from a stochastic model with 100 time steps. We use a Black Scholes model with fixed parameters from Herrera et al. (2021), for which the code is available under the MIT license, in one and two dimension. Visualizations of the data set can be found in Appendix G.

We then subsample timesteps from these trajectories to obtain irregularly sampled timeseries for training. At validation and test time we do not use irregularly but equidistant subsample timesteps for the memory. We show the results for different numbers of subsampling times. We compare our jump-based method, SDE-based method and a Markov superposition of both. As a metric we report the Maximum Mean Discrepancy (MMD) with negative distance kernel Székely & Rizzo (2013) between the generated trajectories and ground truth trajectories using the geomloss Feydy et al. (2019) package. This metric we also use as validation loss. For the architecture, we take a simple feedforward neural network with the Adam optimizer Kingma & Ba (2017). We compare with the TFM method from Zhang et al. (2024) and SDEmatching Bartosh et al. (2025). Hereby we modified the public available code from Bartosh et al. (2025) such that it is applicable to irregular observation points. Further possible baselines would be (latent) NeuralODE/SDE Chen et al. (2018); Rubanova et al. (2019); Li et al. (2020); Kidger et al. (2021b) or aligned Flow Matching Liu et al. (2023a); Somnath et al. (2023). However we did not consider them explicitly here, as those methods were consistently outperformed by TFM in extensive experiments in Zhang et al. (2024). We train each method (Jump/(latent) SDE/TFM) for 4 different subsampling rates using 5 different training seeds each. The choice of the hyperparameters $\eta^2$, $\rho^2$ and, for the Markov superposition, $\alpha$, is done by a parameter search on the validation set, see the supplemental code for implementational details. In case of the two-dimensional data set, we consider three different jump models. For the first version we combine two 1D Gaussian generators $q_t^{(1)}(dy_1, x_1, x_2)$ and $q_t^{(2)}(dy_2, x_1, x_2)$ via factorization,

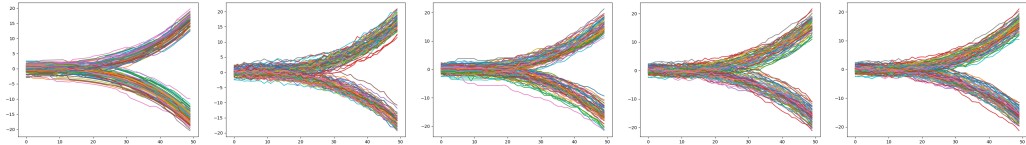

Figure 3: Results on the non-Markovian dataset using a $50\%$ subsampling rate. From Left to Right: Ground truth, TFM, JUMP, SDE method, Markov superposition with $\alpha = 0.90$.

and take as a total loss the sum of the two 1D losses. For the other two versions, we define a Gaussian generator $q_t(\mathrm{d}y_1, \mathrm{d}y_2, x_1, x_2)$ directly in 2D, and employ Proposition 5 and 6 for $d = 2$ to derive the loss. Hereby we use both a fully learnable covariance matrix and a scalar covariance matrix with essentially only one free parameter.

In tables 1, 2 and 3 , we report the performance of methods on a test set (of full times steps) for different subsampling training times. We see that generally, our proposed jump and SDE methods perform much favorably in the subsampled regime, whereas the TFM method Zhang et al. (2024) seems to perform well in the full time steps regime. In particular, the Markov superposition principle greatly improves in most cases, highlighting the usefulness of the generator matching framework. Note that for TFM we stick to the denoising objective proposed in Zhang et al. (2024), whereas for our SDE we predict the velocity. We attribute the improved performance in the subsampled regime also to this choice. The SDE matching Bartosh et al. (2025) results are only reported on the Black–Scholes dataset, since this model is not well suited for capturing non-Markovian behavior (see the extended discussion on memoryless SDE-based models in Zhang et al. (2024)). We provide further details and images in Appendix G.

| Method | 5 | 10 | 25 | 50 |
|---|---|---|---|---|
| Jump-based method | [0.074 ± 0.01] | [0.056 ± 0.02] | [0.078 ± 0.01] | [0.100 ± 0.02] |
| SDE-based method | [**0.041** ± 0.01] | [0.026 ± 0.01] | [**0.033** ± 0.01] | [0.067 ± 0.01] |
| Jump + SDE (Markov superposition) | [**0.040** ± 0.01, $\alpha = 0.9$] | [**0.017** ± 0.002, $\alpha = 0.7$] | [**0.033** ± 0.01, $\alpha = 0.9$] | [0.065 ± 0.01, $\alpha = 0.8$] |
| TFM method Zhang et al. (2024) | [0.593 ± 0.06] | [0.279 ± 0.06] | [0.083 ± 0.05] | [**0.034** ± 0.03] |

Table 1: Average MMD over training runs between generated and ground truth trajectories for different methods and subsampling rates on the synthetic dataset. Memory length $m = 10$ is used.

| Method | 10 | 25 | 50 | 101 |
|---|---|---|---|---|
| Jump-based method | [0.216 ± 0.046] | [0.134 ± 0.025] | [0.134 ± 0.091] | [0.083 ± 0.044] |
| SDE-based method | [0.051 ± 0.003] | [0.054 ± 0.007] | [0.065 ± 0.005] | [0.066 ± 0.005] |
| Jump + SDE (Markov superposition) | [**0.047** ± 0.008, $\alpha = 0.9$] | [**0.035** ± 0.005, $\alpha = 0.8$] | [**0.052** ± 0.020, $\alpha = 0.7$] | [0.064 ± 0.014, $\alpha = 0.9$] |
| TFM method Zhang et al. (2024) | [0.154 ± 0.019] | [0.089 ± 0.017] | [0.166 ± 0.034] | [0.066 ± 0.006] |
| Latent SDE Bartosh et al. (2025) | [0.112 ± 0.019] | [0.101 ± 0.012] | [0.100 ± 0.010] | [**0.033** ± 0.007] |

Table 2: Average MMD over training runs between generated and ground truth trajectories for different methods and subsamplings on the synthetic Black-Scholes dataset in one dimension. Memory length $m = 20$ is used.

| Method | 10 | 25 | 50 | 101 |
|---|---|---|---|---|
| Jump-based method with $\Sigma = \sigma^2 \mathrm{I}$ | [0.443 ± 0.137] | [0.252 ± 0.044] | [0.165 ± 0.020] | [0.192 ± 0.089] |
| Jump-based method with general $\Sigma$ | [0.183 ± 0.083] | [0.108 ± 0.046] | [0.126 ± 0.098] | [0.136 ± 0.073] |
| Jump-based method with factorized components | [0.037 ± 0.005] | [0.066 ± 0.014] | [0.106 ± 0.026] | [**0.103** ± 0.025] |
| SDE-based method | [0.105 ± 0.007] | [0.097 ± 0.005] | [0.107 ± 0.021] | [0.129 ± 0.021] |
| Jump (factorized) + SDE (Markov superposition) | [**0.026** ± 0.004, $\alpha = 0.3$] | [**0.0606** ± 0.022, $\alpha = 0.3$] | [**0.070** ± 0.011, $\alpha = 0.9$] | [**0.103** ± 0.025, $\alpha = 0$] |
| TFM method Zhang et al. (2024) | [0.249 ± 0.029] | [0.255 ± 0.055] | [0.569 ± 0.225] | [0.165 ± 0.042] |

Table 3: Average MMD over training runs between generated and ground truth trajectories for different methods and subsamplings on the synthetic Black-Scholes dataset in two dimensions. Memory length $m = 20$ is used.

## 6.2 VIDEO DATA SET

In order to evaluate the method for high-dimensional data, we introduce a synthetic moving box dataset of 16×16 binary images, where a 3×3 box moves randomly across the grid with boundary reflections and occasional vertical jumps (see Appendix G for details). Hereby, the boxes are initialized randomly and start moving either to the left or to the right for one pixel per step, until they are reflected from the boundary. With a probability of 30 percent in every step, the box jumps upwards or downwards for three pixels. To match training conditions, model outputs are binarized, which may sometimes deviate from a perfect $3 \times 3$ box but still preserve the global dynamics, see Figure 16. In Figure G we show some generated trajectories after training. We quantitatively evaluate how well the dynamics is being learned in Table 4. Here we evaluate how well the (binarized) generated trajectories represent the ground truth dynamics for different combinations of our SDE ($\alpha = 1$) and jump ($\alpha = 0$) model. We observe that the SDE model better learns the 'continuous' horizontal movement, while the jump model better learns the 'discontinuous' vertical movement. Also, the SDE model respects the form of the cube slightly better. The convex combinations are somehow in between the both extremes (except for the first column, where the performance for $\alpha = \{0, 0.2, 0.5\}$ is largely comparable. The TFM performs the best except for the vertical jump ratio, where it is far off.

| $\alpha$ | Well-formed cube | Correct horizontal direction movement | vertical jump ratio | Incorrect vertical jumps |
|---|---|---|---|---|
| 0.0 | $90.70 \pm 4.67$ | $95.53 \pm 2.43$ | $29.30 \pm 4.13$ | $2.41 \pm 0.50$ |
| 0.2 | $90.88 \pm 4.58$ | $96.43 \pm 1.81$ | $30.17 \pm 2.88$ | $2.53 \pm 0.78$ |
| 0.5 | $89.14 \pm 5.53$ | $97.02 \pm 1.54$ | $30.30 \pm 2.27$ | $3.35 \pm 1.32$ |
| 0.8 | $93.65 \pm 2.92$ | $98.40 \pm 0.90$ | $31.96 \pm 1.87$ | $4.94 \pm 1.94$ |
| 1.0 | $96.35 \pm 0.83$ | $98.94 \pm 0.61$ | $32.67 \pm 3.42$ | $5.58 \pm 2.49$ |
| TFM | $100.00 \pm 0.01$ | $99.98 \pm 0.01$ | $38.21 \pm 2.85$ | $2.24 \pm 0.64$ |

Table 4: Quality of the generated box time series. Values are shown as Mean % $\pm$ Standard Deviation computed over 5 seeds with 512 generated trajectories each. Training data vertical jump ratio is 30%.

## 7 LIMITATIONS AND CONCLUSIONS

We introduced trajectory generator matching, applying the generator matching framework to the generation of possibly irregularly sampled time series. To this end, we introduced a stabilized bridge between the data distributions and a derived generator for a corresponding SDE and jump process. We proposed a Gaussian approximation for the jump kernel and derived closed form formulas for the KL divergence with a wide class of jump kernels. In addition, we used a memory heuristic to recover the correct joint distributions. We verified the anaytical formulas and as well the performance of our method on data in various dimension. We limited ourselves to parametrize the rate kernel by Gaussians. The next step would be to employ more complex models, such as k-parametric exponential families or Gaussian mixtures. Lastly, our algorithm could potentially be used to perform uncertainty quantification in price prediction frameworks. In terms of limitations, the Markov superposition and our methods introduce auxiliary hyperparameters which need to be tuned. Further, there is a need for better evaluation metrics of time series data, which would be beneficial for chosing the best model using a validation set.

### REPRODUCIBILITY STATEMENT

We have made several efforts to ensure the reproducibility of our results. A full description of the datasets and their generation procedures is provided in Appendix G. The implementation details of our models and training procedures are described in Section 6. Fixed seeds are reported in the supplementary code. All proofs and theoretical results are included with complete assumptions in the appendix as well. Finally, we submit the code to reproduce all experiments and will make it publically available later.

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

## A   THE USE OF LARGE LANGUAGE MODELS

We used LLMs as an aid to improve our writing. Furthermore we used it to search for related literature and for creating Latex figures and tables.

## B  A ROAD MAP TO TIME SERIES WITH JUMP PROCESSES

This section provides an accessible, high-level overview of how generator matching can be used to model time series with jumps. The goal is to give intuition for the construction, complementing the formal development in the main text.

**Task.**  We are given $N$ time series $\{x^1, \ldots, x^N\}$, each consisting of a set of obervations $x^i = (x^i_{t^i_1}, \ldots, x^i_{t^i_{n_i}})$ at irregular and varying times. Our objective is to learn a continuous time stochastic process whose finite-dimensional marginal law at the observation times reproduce the empirical time series distribution.

**Jump processes in a nutshell.**  A pure jump process is determined by two objects:

- a *jump kernel* $J_t(\cdot, x)$ describing where the process proposes to jump when located at $x$ at time $t$,
- a *jump intensity* $\lambda_t(x)$ specifying how frequently such jumps occur.

Intuitively, $J_t$ draws a candidate future position and $\lambda_t$ decides whether the trajectory "stays" or "teleports" there. Points that are unlikely under the data distribution should have large $\lambda_t(x)$, pushing the process away.

**Anchoring the dynamics via analytical bridges.**  To obtain a tractable learning problem, we start from an analytically solvable reference family. Between two points $x_0 = x^i_{t_j}$ and $x_1 = x^i_{t_{j+1}}$ at $t_0 = t^i_{t_j}$ and $t_1 = t^i_{t_{j+1}}$, Proposition 2 provides closed-form formulas for a drift–diffusion bridge and a jump bridge whose marginals are Gaussian

$$P_t = \mathcal{N}(m_t, \tau_t), \qquad m_t = \frac{t_1 - t}{t_1 - t_0} x_0 + \frac{t - t_0}{t_1 - t_0} x_1, \quad \tau_t = \eta^2 \frac{(t - t_0)(t_1 - t)}{(t_1 - t_0)^2} + \rho^2.$$

These formulas yield explicit candidate jump intensities and jump kernels but they depend on the chosen data points.

**From analytical bridges to learning via conditional losses.**  Following the spirit of flow matching, the analytical generator formulas tell us what the correct infinitesimal drift and jump kernels would be *if we knew the endpoints*. Using disintegration (Appendix C), one shows that regressing appropriate conditional versions of these formulas yields a learned generator whose marginal laws match the desired conditionals.

The jump part requires special care because $J_t$ is a whole probability distribution, not a vector. Using the raw formula from Proposition 2 would require reverse KL, binning, or other potentially unstable approximations. To avoid this, we restrict the neural network to output a *Gaussian* jump kernel. This choice leads to:

- a closed-form expression of the KL loss (Proposition 6),
- explicit closed-form formulas for the mean and covariance of the target kernel in Proposition 6.

This makes the training stable and computationally efficient.

**Why the method is flexible.**  The framework enjoys several attractive properties:

- *Irregular and varying time series can be processed.* The generator is defined in continuos time and thus can be applied to arbitrary, irregular and varying observation times.
- *Jump and diffusion dynamics can be combined* through the superposition principle, enabling expressive mixed dynamics.
- *Non-Markovian time series can be handled* by conditioning the generator on a memory window of recent observations. This introduces path-dependence while keeping the conditional generator analytically tractable.

Together, these components explain conceptually how the formal construction in the main manuscript leads to a practically usable generative model for irregular time series with jumps.

## C ADDITIONAL DEFINITIONS

Let $\mathcal{M}_+^{\mathrm{ac}}(\mathbb{R}^d)$ denote the space of absolutely continuous non-negative measures on $\mathbb{R}^d$ with respect to the Lebesgue measure. Our loss function for the jump process relies on the *Kullback-Leibler divergence* $D_{\mathrm{KL}} : \mathcal{M}_+^{\mathrm{ac}}(\mathbb{R}^d) \times \mathcal{M}_+^{\mathrm{ac}}(\mathbb{R}^d) \to [0, \infty]$ defined by

$$D_{\mathrm{KL}}(p, q) := \int p(x) \log\left(\frac{p(x)}{q(x)}\right) - p(x) + q(x) \, \mathrm{d}x.$$

if $p(x) = 0$ whenever $q(x) = 0$ for a.e. $x \in \mathbb{R}^d$, and $D_{\mathrm{KL}}(p, q) := \infty$ otherwise. We have that $D_{\mathrm{KL}}(p, q) = 0$ if and only if $p = q$ a.e..

For a measure $\alpha \in \mathcal{P}(\mathbb{R}^d \times \mathbb{R}^d)$ and projections $\pi^x, \pi^y : \mathbb{R}^d \times \mathbb{R}^d \to \mathbb{R}^d$ defined by

$$\pi^x(x, y) := x, \quad \pi^y(x, y) := y$$

we have that $\pi_\sharp^x \alpha$ and $\pi_\sharp^y \alpha$ are the left and right marginals of $\alpha$, respectively. Then for $\alpha \in \mathcal{P}(\mathbb{R}^d \times \mathbb{R}^d)$ with marginal $\pi_\sharp^x \alpha = \mu$, there exists a $\mu$-a.e. uniquely defined family of probability measures $\{\alpha_x\}_x$, called *disintegration of $\alpha$ with respect to $\pi^x$*, such that the map $x \mapsto \alpha_x(B)$ is measurable for every $B \in \mathcal{B}(\mathbb{R}^d)$, and

$$\alpha = \alpha_x \times_x \mu$$

meaning that

$$\int_{\mathbb{R}^d \times \mathbb{R}^d} f(x, y) \, \mathrm{d}\alpha(x, y) = \int_{\mathbb{R}^d} \int_{\mathbb{R}^d} f(x, y) \, \mathrm{d}\alpha_x(y) \, \mathrm{d}\mu(x)$$

for every measurable, bounded function $f : \mathbb{R}^d \times \mathbb{R}^d \to \mathbb{R}$. Similarly, we define for a measure $\alpha \in \mathcal{P}(\mathbb{R}^d \times \mathbb{R}^d)$ with marginal $\pi_\sharp^y \alpha = \nu$ the *disintegration of $\alpha$ with respect to $\pi^y$* as

$$\alpha = \alpha_y \times_y \nu.$$

The notation of disintegration is directly related to Markov kernels. A *Markov kernel* is a map $\mathcal{K} : \mathbb{R}^d \times \mathcal{B}(\mathbb{R}^d) \to \mathbb{R}$ such that

    i) $\mathcal{K}(x, \cdot)$ is a probability measure on $\mathbb{R}^d$ for every $x \in \mathbb{R}^d$, and

    ii) $\mathcal{K}(\cdot, B)$ is a Borel measurable map for every $B \in \mathcal{B}(\mathbb{R}^d)$.

Hence, given a probability measure $\mu \in \mathcal{P}(\mathbb{R}^d)$, we can define a new measure $\alpha := \alpha_x \times_x \mu \in \mathcal{P}(\mathbb{R}^d \times \mathbb{R}^d)$ by

$$\int f(x, y) \, \mathrm{d}\alpha(x, y) := \int \int f(x, y), \, \mathrm{d}\mathcal{K}(x, \cdot)(y) \, \mathrm{d}\mu(x)$$

for all measurable, bounded functions $f$. Identifying $\alpha_x(B)$ with $\mathcal{K}(x, B)$, we see that conversely, $\{\alpha_x\}_x$ is the disintegration of $\alpha$ with respect to $\pi^1$. Let $X_0, X_1 : \Omega \to \mathbb{R}^d$ be random variables with joint distribution $P_{X_0, X_1}$. Then the conditional distribution $\{P_{X_1 | X_0 = x}\}_x$ provides the disintegration of $P_{X_0, X_1}$, i.e.,

$$\int f(x, y) \, \mathrm{d}P_{X_0, X_1}(x, y) = \int \int f(x, y) \, \mathrm{d}P_{X_1 | X_0 = x}(y) \, \mathrm{d}P_{X_0}(x).$$

In other words, $\mathcal{K} = P_{X_1 | X_0}$ is a Markov kernel. If $P_{X_0, X_1}$ admits a density $p_{X_0, X_1}$ and $P_{X_0}$ a density $p_{X_0} > 0$, then $P_{X_1 | X_0 = x}$ has the density $p_{X_1 | X_0 = x} = p_{X_0, X_1}(x, \cdot)/p_{X_0}(x)$.

## D PROOFS

**Proof of Proposition 2** We consider more generally the interpolation points $t_0$ and $t_1$ instead of 0 and 1, and get the following expressions:

- $m_t := \frac{t_1 - t}{t_1 - t_0} x_0 + \frac{t - t_0}{t_1 - t_0} x_1$ and $\tau_t := \eta^2 \frac{(t - t_0)(t_1 - t)}{(t_1 - t_0)^2} + \rho^2,$

- $u_t := \frac{x_1 - x_0}{t_1 - t_0} + (x - m_t) \frac{\eta^2}{2\tau_t} \left( \frac{t_0 + t_1 - 2t}{(t_1 - t_0)^2} - 1 \right),$

- $a_t := \frac{\sqrt{\tau_t}(t_1 - t_0)}{\eta^2(t_0 + t_1 - 2t)}(x_1 - x_0)$ and $\xi_t(y) := \frac{\eta^2(t_0 + t_1 - 2t)}{2(t_1 - t_0)^2 \tau_t} \left( \frac{\|y - m_t\|^2}{\tau_t} + 2a_t^T \frac{y - m_t}{\sqrt{\tau_t}} - d \right).$

For $t = t_0$, we see that $p_{t_0} \sim \mathcal{N}(x_0, \rho^2)$ and for $t = t_1$ that $p_{t_1} \sim \mathcal{N}(x_1, \rho^2)$. We have $m_t' = \frac{x_1 - x_0}{t_1 - t_0}$ and $\tau_t' = \frac{\eta^2}{(t_1 - t_0)^2}(t_0 + t_1 - 2t)$. Computing the time derivative of $p_t$ yields

$$\partial_t p_t(x) = p_t(x) \left( - \frac{d \, \tau_t'}{2\tau_t} + \frac{(x_1 - x_0)^{\mathsf{T}}(x - m_t)}{\tau_t(t_1 - t_0)} + \frac{\|x - m_t\|^2 \tau_t'/2}{\tau_t^2} \right)$$

$$= p_t(x) \left( - \frac{d \, \eta^2(t_0 + t_1 - 2t)}{2\tau_t(t_1 - t_0)^2} + \frac{(x_1 - x_0)^{\mathsf{T}}(x - m_t)}{\tau_t(t_1 - t_0)} + \frac{\|x - m_t\|^2 \eta^2(t_0 + t_1 - 2t)}{2\tau_t^2(t_1 - t_0)^2} \right)$$

$$= p_t(x)\xi_t(x).$$

First, we deal with the diffusion part. Using that $\nabla \cdot u_t = \frac{d \, \eta^2}{2\tau_t} \left( \frac{t_0 + t_1 - 2t}{(t_1 - t_0)^2} - 1 \right)$ and $\nabla p_t = -p_t \frac{x - m_t}{\tau_t}$, we can compute

$$\nabla \cdot (p_t u_t)(x) = u_t^{\mathsf{T}} \nabla p_t + p_t \nabla \cdot u_t = -p_t \left( \frac{u_t^{\mathsf{T}}(x - m_t)}{\tau_t} - \frac{d \, \eta^2}{2\tau_t} \left( \frac{t_0 + t_1 - 2t}{(t_1 - t_0)^2} - 1 \right) \right)$$

$$= -p_t \left( \frac{(x_1 - x_0)^{\mathsf{T}}(x - m_t)}{\tau_t(t_1 - t_0)} + \|x - m_t\|^2 \frac{\eta^2}{2\tau_t^2} \left( \frac{t_0 + t_1 - 2t}{(t_1 - t_0)^2} - 1 \right) \right.$$

$$\left. - \frac{d \, \eta^2}{2\tau_t} \left( \frac{t_0 + t_1 - 2t}{(t_1 - t_0)^2} - 1 \right) \right).$$

Further, we obtain

$$\Delta p_t(x) = -\nabla p_t^{\mathsf{T}} \frac{x - m_t}{\tau_t} - \frac{d \, p_t}{\tau_t} = p_t \left( \frac{\|x - m_t\|^2}{\tau_t^2} - \frac{d}{\tau_t} \right)$$

so that

$$-\nabla \cdot (p_t u_t) + \frac{1}{2}\eta^2 \Delta p_t = p_t \left( \frac{(x_1 - x_0)^{\mathsf{T}}(x - m_t)}{\tau_t(t_1 - t_0)} + \|x - m_t\|^2 \frac{\eta^2}{2\tau_t^2} \left( \frac{t_0 + t_1 - 2t}{(t_1 - t_0)^2} - 1 \right) \right.$$

$$\left. - \frac{d \, \eta^2}{2\tau_t} \left( \frac{t_0 + t_1 - 2t}{(t_1 - t_0)^2} - 1 \right) + \frac{\eta^2}{2} \left( \frac{\|x - m_t\|^2}{\tau_t^2} - \frac{d}{\tau_t} \right) \right)$$

$$= \left( \frac{\eta^2}{2\tau_t} \left( -d \left( \frac{t_0 + t_1 - 2t}{(t_1 - t_0)^2} - 1 \right) - d \right) + \frac{(x_1 - x_0)^{\mathsf{T}}(x - m_t)}{\tau_t(t_1 - t_0)} \right.$$

$$\left. + \|x - m_t\|^2 \frac{\eta^2}{2\tau_t^2} \left( \frac{t_0 + t_1 - 2t}{(t_1 - t_0)^2} - 1 + 1 \right) \right)$$

$$= \partial_t p_t(x).$$

For the jump part, we get

$$\int p_t(y)q_t(x, y) - p_t(x)q_t(y, x)\mathrm{d}y = J_t(x) \int \lambda_t(y)p_t(y)\mathrm{d}y - \lambda_t(x)p_t(x) \tag{6}$$

Since $\max(0, \xi_t) = \xi_t + \lambda_t$, we conclude by definition of $\xi_t$ that

$$\int \max(0, \xi_t)p_t \, \mathrm{d}y = \int \xi_t p_t \, \mathrm{d}y + \int \lambda_t p_t \, \mathrm{d}y = \mathbb{E}_{y \sim \mathcal{N}(m_t, \tau_t)}[\xi_t] + \int \lambda_t p_t \, \mathrm{d}y$$

$$= \frac{\eta^2(t_0 + t_1 - 2t)}{2(t_1 - t_0)^2 \tau_t}(d + 0 - d) + \int \lambda_t p_t \, \mathrm{d}y = \int \lambda_t p_t \, \mathrm{d}y.$$

Hence we can rewrite

$$J_t(x) = \frac{(\xi_t(x) + \lambda_t(x))p_t(x)}{\int \lambda_t(y)p_t \, \mathrm{d}y}, \tag{7}$$

which is well-defined as long as $t \neq (t_0 + t_1)/2$. If $t = (t_0 + t_1)/2$, then $\lambda_t = 0$, and we can set $J_t = 0$. Using (7) in (6), we obtain finally

$$\int p_t(y)q_t(x,y) - p_t(x)q_t(y,x)\,\mathrm{d}y = p_t(x)\xi_t(x) = \partial_t p_t(x).$$

This finishes the proof. □

**Proof of Remark 3.** We have that $Y_t^{x_0} \sim P_t^{x_0}$. Since

$$\int_{\mathbb{R}^d} f(x)\,\mathrm{d}P_0^{x_0}(x) = \int_{\mathbb{R}^d \times \mathbb{R}^d} f(x)\,\mathrm{d}P_0(\,\mathrm{d}x, x_0, x_1)\,\mathrm{d}P_{X_1|X_0=x_0}$$

$$= \int f(x)\int \mathrm{d}\mathcal{N}(x_0, \rho^2)\,\mathrm{d}P_{X_1|X_0=x_0}(x_1) = \int_{\mathbb{R}^d} f(x)\,\mathrm{d}\mathcal{N}(x_0, \rho^2)$$

$$\int_{\mathbb{R}^d} f(x)\,\mathrm{d}P_1^{x_0}(x) = \int_{\mathbb{R}^d \times \mathbb{R}^d} f(x)\,\mathrm{d}\mathcal{N}(x_1, \rho^2)(x)\,\mathrm{d}P_{X_1|X_0=x_0}$$

$$= \int_{\mathbb{R}^d} f(x)\left(\int_{\mathbb{R}^d} \frac{1}{\rho\sqrt{2\pi}} e^{-\frac{\|x-x_1\|^2}{2\rho^2}}\,\mathrm{d}P_{X_1|X_0=x_0}(x_1)\right)\,\mathrm{d}x$$

$$= \int f(x)\,\mathrm{d}P_{X_1|X_0=x_0} * \mathcal{N}(0, \rho^2)$$

we obtain distributions for $Y_0^{x_0}, Y_1^{x_0}$. □

**Proof of Proposition 5.** By definition of the KL-divergence, we get

$$F_{t,x}(\lambda, \mu, \Sigma) = \lambda - \lambda_t(x) + \int q_t(y,x)\log\left(\frac{q_t(y,x)}{\lambda\mathcal{N}(\mu, \Sigma)(y)}\right)\,\mathrm{d}y$$

$$= \lambda - \lambda_t(x) + \int q_t(y,x)\left(\log(q_t(y,x)) - \log(\lambda)\right)$$

$$+ \int q_t(y,x)\left(\frac{d}{2}\log(2\pi) + \frac{1}{2}\log(\det \Sigma) + \frac{1}{2}\langle y - \mu, \Sigma^{-1}(y - \mu)\rangle\right)\,\mathrm{d}y$$

$$= \lambda + \lambda_t(x)\left(\frac{d}{2}\log(2\pi) + \frac{1}{2}\log(\det \Sigma) - \log(\lambda) + \frac{1}{2}\int J_t(y)\langle y - \mu, \Sigma^{-1}(y - \mu)\rangle\,\mathrm{d}y\right)$$

$$+ \int q_t(y,x)\log(q_t(y,x))\,\mathrm{d}y - \lambda_t(x)$$

$$= \lambda + \lambda_t(x)\left(\frac{1}{2}\log(\det \Sigma) - \log(\lambda) + \frac{1}{2}\langle \Sigma^{-1}, \Sigma^J\rangle + \frac{1}{2}\langle \mu^J - \mu, \Sigma^{-1}(\mu^J - \mu)\rangle\right) + \text{const.}$$

Here, the identity $\int J_t(y)\langle y - \mu, \Sigma^{-1}(y - \mu)\rangle\,\mathrm{d}y = \frac{1}{2}\langle \Sigma^{-1}, \Sigma^J\rangle + \frac{1}{2}\langle \mu^J - \mu, \Sigma^{-1}(\mu^J - \mu)\rangle$ follows using the decomposition $y - \mu = (y - \mu^J) + (\mu^J - \mu)$. For $\Sigma = \sigma^2 \mathrm{Id}_d$ setting the partial derivatives of $F_{t,x}$ to zero leads to

$$\partial_\mu F_{t,x}(\lambda, \mu, \sigma) = \lambda_t(x)\frac{\mu - \mu^J}{\sigma^2} = 0,$$

$$\partial_\sigma F_{t,x}(\lambda, \mu, \sigma) = \frac{\lambda_t(x)}{\sigma^3}\left(d\sigma^2 - \text{trace }\Sigma^J - \|\mu^J - \mu\|^2\right) = 0,$$

$$\partial_\lambda F_{t,x}(\lambda, \mu, \sigma) = \frac{\lambda - \lambda_t(x)}{\lambda} = 0,$$

and the solution $(\lambda, \mu, \sigma) = (\lambda_t(x), \mu^J, \sqrt{\frac{1}{d}\text{trace }\Sigma^J})$ is the unique critical point of $F_{t,x}$. This is the global minimum, since the Hessian of $F_{t,x}$ at this point is $\lambda_t(x)\text{diag}\left(\frac{1}{\sigma^2}, \frac{2}{\sigma^2}, \frac{1}{\lambda^2}\right)$ which is positive definite for $\lambda_t(x) > 0$. □

### D.1 PROOF OF PROPOSITION 6.

We formulate the statement separately for both cases: We start with the case $d = 1$.

**Proposition 8.** *For dimension $d = 1$, let $\mu^J = \mu_{x_0,x_1,J}$ and $(\sigma^J)^2 = (\sigma^{x_0,x_1,J})^2$ denote the mean and variance of $J_t$. For arbitrary $t_0$ and $t_1$ define*

- $m_t := \frac{t_1 - t}{t_1 - t_0} x_0 + \frac{t - t_0}{t_1 - t_0} x_1$ *and* $\tau_t := \eta^2 \frac{(t - t_0)(t_1 - t)}{(t_1 - t_0)^2} + \rho^2$,

- $u_t := \frac{x_1 - x_0}{t_1 - t_0} + (x - m_t) \frac{\eta^2}{2\tau_t} \left( \frac{t_0 + t_1 - 2t}{(t_1 - t_0)^2} - 1 \right)$,

- $a_t := \frac{\sqrt{\tau_t}(t_1 - t_0)}{\eta^2(t_0 + t_1 - 2t)}(x_1 - x_0)$ *and* $\xi_t(y) := \frac{\eta^2(t_0 + t_1 - 2t)}{2(t_1 - t_0)^2 \tau_t} \left( \frac{\|y - m_t\|^2}{\tau_t} + 2a_t^T \frac{y - m_t}{\sqrt{\tau_t}} - d \right)$.

*Let*

$$z_\pm := -a_t \pm \sqrt{a_t^2 + 1}.$$

*Further, define*

$$I_0 := \sqrt{\pi/2}\left( \mathrm{erf}\left( \frac{z_+}{\sqrt{2}} \right) - \mathrm{erf}\left( \frac{z_-}{\sqrt{2}} \right) \right), \quad I_1 := e^{-z_-^2/2} - e^{-z_+^2/2},$$

$$I_k = (k-1)I_{k-2} - \left( z_+^{k-1} e^{-z_+^2/2} - z_-^{k-1} e^{-z_-^2/2} \right), \quad k = 2, 3, 4.$$

*Then it holds for $t \in \left( \frac{t_0 + t_1}{2}, t_1 \right]$ that*

$$\mu^J = m_t + \sqrt{\tau_t} \frac{I_3 + 2a_t I_2 - I_1}{I_2 + 2a_t I_1 - I_0}, \tag{8}$$

$$(\sigma^J)^2 = \tau_t \frac{I_4 + 2a_t I_3 - I_2}{I_2 + 2a_t I_1 - I_0} - (m_t - \mu^J)^2.$$

*For $t \in \left[ t_0, \frac{t_0 + t_1}{2} \right)$, the values $I_k$ have to be replaced by $\frac{(-1)^k + 1}{2} \sqrt{2\pi}(k-1)!! - I_k$, $k = 0, \ldots, 4$.*

Setting $z := (y - m_t)/\sqrt{\tau_t}$, we obtain

$$\xi_t(z) = \frac{\eta^2(t_0 + t_1 - 2t)}{2\tau_t(t_1 - t_0)^2} \left( z^2 + 2a_t z - 1 \right).$$

This quadratic polynomial has zeros

$$z_\pm = -a_t \pm \sqrt{a_t^2 + 1}.$$

For $t \in [t_0, \frac{t_0 + t_1}{2})$ the support of $J_t(z)$ is $(-\infty, z_-] \cup [z_+, \infty)$, for $t \in (\frac{t_0 + t_1}{2}, t_1]$ it is $[z_-, z_+]$. Substituting

$$C_t := \int_{z_-}^{z_+} (z^2 + 2a_t z - 1)\, e^{-z^2/2}\, \mathrm{d}z,$$

we conclude for $t \in (\frac{t_0 + t_1}{2}, t_1]$ that the mean is

$$\mu^J = \int y J_t(y)\, \mathrm{d}y = \frac{1}{C_t} \int_{z_-}^{z_+} (z\sqrt{\tau_t} + m_t)(z^2 + 2a_t z - 1)\, e^{-z^2/2}\, \mathrm{d}z$$

$$= m_t + \frac{\sqrt{\tau_t}}{C_t} \int_{z_-}^{z_+} z\,(z^2 + 2a_t z - 1)\, e^{-z^2/2}\, \mathrm{d}z \tag{9}$$

and the variance

$$(\sigma^J)^2 = \int (y - (\mu^J))^2 J_t(y)\, \mathrm{d}y = \int y^2 J_t(y)\, \mathrm{d}y - (\mu^J)^2 \tag{10}$$

$$= \frac{1}{C_t} \int_{z_-}^{z_+} (\sqrt{\tau_t} z + m_t)^2 (z^2 + 2a_t z - 1)\, e^{-z^2/2}\, \mathrm{d}z - (\mu^J)^2.$$

Thus, we need expressions for

$$I_k := \int_{z_-}^{z_+} z^k e^{-z^2/2}\, \mathrm{d}z, \quad k = 0, \ldots, 4.$$

The value $I_0$ follows by the definition of the Gaussian error function

$$\int_{-\infty}^{z} e^{-z^2/2}/\sqrt{2\pi}\, \mathrm{d}z = \frac{1}{2}\left(1 + \mathrm{erf}(z/\sqrt{2})\right).$$

For $k \geq 1$, and with a slight abuse of notation for $k = 1$, integration by parts gives

$$\int_{z_-}^{z_+} z^k \mathrm{e}^{-z^2/2}\, \mathrm{d}z = -z^{k-1}\mathrm{e}^{-z^2/2}|_{z_-}^{z_+} + (k-1)\int_{z_-}^{z_+} z^{k-2}\mathrm{e}^{-z^2/2}\, \mathrm{d}z,$$

so that

$$I_k = (k-1)I_{k-2} - \left(z_+^{k-1}\mathrm{e}^{-z_+^2/2} - z_-^{k-1}\mathrm{e}^{-z_-^2/2}\right).$$

Plugging this into (9) and (10), for $t \in (\frac{t_0+t_1}{2}, t_1]$, we obtain finally

$$\mu^J = m_t + \sqrt{\tau_t}\frac{I_3 + 2a_t I_2 - I_1}{I_2 + 2a_t I_1 - I_0},$$

$$(\sigma^J)^2 = \frac{1}{C_t}\left(\tau_t \int_{z_-}^{z_+} z^2(z^2 + 2a_t z - 1)\, \mathrm{d}z + 2\sqrt{\tau_t}m_t \int_{z_-}^{z_+} z(z^2 + 2a_t z - 1)\, \mathrm{d}z\right) + m_t^2 - (\mu^J)^2$$

$$= \frac{\tau_t}{C_t}\int_{z_-}^{z_+} z^2(z^2 + 2a_t z - 1)\, \mathrm{d}z + 2m_t(\mu^J - m_t) + m_t^2 - (\mu^J)^2$$

$$= \tau_t \frac{I_4 + 2a_t I_3 - I_2}{I_2 + 2a_t I_1 - I_0} - (m_t - \mu^J)^2.$$

For $t \in [t_0, \frac{t_0+t_1}{2})$, and using the well-known formular for Gaussian moments

$$\int_{-\infty}^{\infty} z^k e^{-z^2/2}\mathrm{d}z/\sqrt{2\pi} = \begin{cases} 0 & k \text{ odd} \\ (k-1)!! & k \text{ even} \end{cases},$$

we obtain the above values, where the $I_k$ have to be replaced with $\frac{(-1)^k+1}{2}\sqrt{2\pi}(k-1)!! - I_k$. $\qquad\square$

**Proposition 9.** *Let $d = 2$ and $t \neq \frac{t_0+t_1}{2}$. Define $r_\pm(a_t) = \sqrt{\|a_t\|^2 + d} \pm \|a_t\|$ and for $r \in (r_-(a_t), r_+(a_t))$ the angle $\phi(r, a_t) = \arccos(\frac{d-r^2}{2\|a_t\|r})$. Set*

$$f_0(r, a_t) = \begin{cases} 2\pi(r^2 - d) & r \geq r_+(a_t) \\ 2\phi(r, a_t) + 4\|a_t\|r\sin(\phi(r, a_t)) & r \in (r_-(a_t), r_+(a_t)) \\ 0 & r \leq r_-(a_t) \end{cases}$$

$$f_1(r, a_t) = \begin{cases} 2\pi\|a_t\|r & r \geq r_+(a_t) \\ 2(r^2 - d)\sin(\phi(r, a_t)) & \\ \quad +2\|a_t\|r\left(\phi(r, a_t) + \frac{1}{2}\sin(2\phi(r, a_t))\right) & r \in (r_-(a_t), r_+(a_t)) \\ 0 & r \leq r_-(a_t) \end{cases}$$

$$f_2(r, a_t) = \begin{cases} \pi(r^2 - d) & r \geq r_+(a_t) \\ (r^2 - d)\left(\phi(r, a_t) + \frac{1}{2}\sin(2\phi(r, a_t))\right) & \\ \quad +2\|a_t\|r\left(\frac{3}{2}\sin(\phi(r, a_t)) + \frac{1}{6}\sin(3\phi(r, a_t))\right) & r \in (r_-(a_t), r_+(a_t)) \\ 0 & r \leq r_-(a_t) \end{cases}$$

*Then, we have that*

$$\mu^J = E[J_t] = m_t + \sqrt{\tau_t}\frac{a_t}{\|a_t\|}\frac{\int_0^\infty e^{-\frac{r^2}{2}}r^2 f_1(r, a_t)\mathrm{d}r}{\int_0^\infty e^{-\frac{r^2}{2}}r f_0(r, a_t)\mathrm{d}r}, \tag{11}$$

$$\mathrm{Cov}(J_t) = \tau_t \frac{a_t a_t^T}{\|a_t\|^2}\frac{\int_0^\infty e^{-\frac{r^2}{2}}r^3 f_2(r, a_t)\mathrm{d}r}{\int_0^\infty e^{-\frac{r^2}{2}}r f_0(r, a_t)\mathrm{d}r} - (\mu^J - m_t)(\mu^J - m_t)^T.$$

*Proof.* We can proceed similarly to Proposition 8. Everything reduces to the computation of the moments $\int_{\mathbb{R}^d} \max(0, \xi_t(y)) p_t(y) \mathrm{d}y$, $\int_{\mathbb{R}^d} y \max(0, \xi_t(y)) p_t(y) \mathrm{d}y$ and $\int_{\mathbb{R}^d} y^2 \max(0, \xi_t(y)) p_t(y) \mathrm{d}y$. Let $R_t \in \mathbb{R}^{d \times d}$ be an orthogonal matrix with $\det(R_t) = 1$ and $R_t a_t = \|a_t\| e_1$, where $e_1 \in \mathbb{R}^d$ is the first cartesian unit vector in $\mathbb{R}^d$. The change of coordinates given by the substitution $v = R_t \left( \frac{y - m_t}{\sqrt{\tau_t}} \right) =: S_t(y)$ simplifies $J_t$ in the sense that it depends only on $\|v\|$ and $v_1$. Concretely, it yields

$$\max(0, \xi_t(y)) p_t(y) \mathrm{d}y = \text{const} \cdot \max\left(0, \|v\|^2 + 2\|a_t\| v_1 - d\right) \cdot e^{-\|v\|^2/2} \tau_t^{d/2} \mathrm{d}v.$$

Employing this coordinate transformation and switching to polar coordinates in the case $d = 2$, the moments reduce to radial integrals, which yields the expressions above. $\square$

## E  PROOF OF PROPOSITION 7

In order to show the statement we proceed in two steps. First, we show that we can approximate $P_{X_0, X_1}$ if we can approximate $P_{X_0}$ and $P_{X_1 | X_0 = x_0}$ by $P_{Y_0}$ and a Markov kernel $\mu(\cdot, x_0)$ respectively.

**Lemma 10.** *Let $P_{X_0, X_1} \in \mathcal{P}_2(\mathbb{R}^m \times \mathbb{R}^d)$, $P_{Y_0} \in \mathcal{P}_2(\mathbb{R}^m)$ and let $\mu(\cdot, x_0) : \mathcal{B}(\mathbb{R}^d) \times \mathbb{R}^m \to [0, 1]$ be a Markov kernel. Define $\alpha := \mu(\cdot, x_0) \times_{x_0} P_{Y_0} \in \mathcal{P}_2(\mathbb{R}^m \times \mathbb{R}^d)$. Assume that*

i) $\int W_2^2(P_{X_1 | X_0 = a}, \mu(\cdot, x_0)) \, \mathrm{d}P_{X_0}(a) \leq \epsilon$,

ii) $W_2(P_{X_1 | X_0 = a}, P_{X_1 | X_0 = b}) \leq K \|a - b\|$,

iii) $W_2(\mu(\cdot, a), \mu(\cdot, b)) \leq K \|a - b\|$.

*Then we have*

$$W_2(P_{X_0, X_1}, \alpha)^2 \leq (1 + 20\,K^2) W_2^2(P_{X_0}, P_{Y_0}) + 4\epsilon.$$

*Proof.* Let $\gamma_{a,b} \in \Gamma_o(P_{X_1 | X_0 = a}, \mu(\cdot, b))$ and let $\beta \in \Gamma_o(P_{X_0}, P_{Y_0})$. First note that $\gamma_{a,b} \times_{a,b} \beta \in \Gamma(P_{X_1, X_2}, \alpha)$. Then

$$W_2^2(P_{X_0, X_1}, \alpha) \leq \int \|(a, x) - (b, y)\|^2 \, \mathrm{d}\gamma_{a,b}(x, y) \, \mathrm{d}\beta(a, b)$$

$$= \int \|a - b\|^2 \, \mathrm{d}\beta + \int \|x - y\|^2 \, \mathrm{d}\gamma_{a,b}(x, y) \, \mathrm{d}\beta$$

$$= W_2^2(P_{X_0}, P_{Y_0}) + \int W_2^2\left(P_{X_2 | X_1 = a}, \mu(\cdot, b)\right) \, \mathrm{d}\beta$$

$$\leq W_2^2(P_{X_0}, P_{Y_0}) + \int \left(W_2(P_{X_1 | X_0 = a}, P_{X_1 | X_0 = b}) + W_2(P_{X_1 | X_0 = b}, \mu(\cdot, b))\right)^2 \, \mathrm{d}\beta$$

$$\leq W_2^2(P_{X_0}, P_{Y_0}) + 2 \int W_2^2(P_{X_1 | X_0 = a}, P_{X_1 | X_0 = b}) + W_2^2(P_{X_1 | X_0 = b}, \mu(\cdot, b)) \, \mathrm{d}\beta$$

$$\leq (1 + 2\,K^2) W_2^2(P_{X_0}, P_{Y_0}) + 2 \int W_2^2(P_{X_1 | X_0 = b}, \mu(\cdot, b)) \, \mathrm{d}\beta$$

$$\leq (1 + 2\,K^2) W_2^2(P_{X_0}, P_{Y_0})$$
$$+ 2 \int \left(W_2(P_{X_1 | X_0 = b}, P_{X_1 | X_0 = a}) + W_2(P_{X_1 | X_0 = a}, \mu(\cdot, a)) + W_2(\mu(\cdot, a), \mu(\cdot, a))\right)^2 \, \mathrm{d}\beta$$

$$\leq (1 + 2\,K^2) W_2^2(P_{X_0}, P_{Y_0})$$
$$+ 8 \int W_2^2(P_{X_1 | X_0 = b}, P_{X_1 | X_0 = a}) + 4\,W_2^2(P_{X_1 | X_0 = a}, \mu(\cdot, a)) + 8\,W_2^2(\mu(\cdot, a), \mu(\cdot, b)) \, \mathrm{d}\beta$$

$$\leq (1 + 20\,K^2) W_2^2(P_{X_0}, P_{Y_0}) + 4\epsilon.$$

$\square$

This result can be easily extended to several time points as follows.

**Proposition 11.** *Let* $X_0, \dots, X_n, Y_0 \in L^2(\Omega, \mathbb{P})$ *and let* $\mu_i(\cdot, x_0, \dots, x_i)$ *be Markov kernels in* $x_0, \dots, x_i$. *Let* $\mu := \mu(\cdot, x_0, \dots, x_{n-1}) \times_{x_0, \dots, x_{n-1}} \cdots \times_{x_0} P_{Y_0} \in \mathcal{P}_2(\mathbb{R}^{(n+1)d})$ *such that*

*i)* $\int W_2^2(P_{X_{i+1}|X_0=a_0, \dots X_i=a_i}, \mu(\cdot, a_0, \dots, a_i) \, \mathrm{d}P_{X_0, \dots, X_i}(a_0, \dots, a_i) \le \epsilon,$

*ii)* $W_2\left(P_{X_{i+1}|X_0=a_0^1, \dots, X_i=a_i^1}, P_{X_{i+1}|X_0=a_0^2, \dots, X_i=a_i^2}\right) \le K\|(a_0^1, \dots, a_i^1) - (a_0^2, \dots, a_i^2)\|,$

*iii)* $W_2\left(\mu(\cdot, b_0^1, \dots, b_i^1), \mu(\cdot, b_0^2, \dots, b_i^2)\right) \le K\|(b_0^1, \dots, b_i^1) - (b_0^2, \dots, b_i^2)\|.$

*Then it holds*

$$W_2^2(P_{X_0, \dots, X_n}, \mu) \le 4\epsilon \sum_{i=0}^{n-1} \left(1 + 20K^2\right)^i + (1 + 20K^2)^n W_2^2(P_{X_0}, P_{Y_0}).$$

*Proof.* This follows via induction from Lemma 10. $\square$

Note that in the situation of section 5 we do not approximate the posterior but $P_{X_{i+1}|X_0=x_0, \dots, X_i=x_i} * \mathcal{N}(0, \rho^2)$. Since by (Ambrosio et al., 2008, Lemma 7.1.10) we have that $W_2(P_{X_{i+1}|X_0=x_0, \dots, X_i=x_i}, P_{X_{i+1}|X_0=x_0, \dots, X_i=x_i} * \mathcal{N}(0, \rho^2)) \le \rho^2$ we could use $\epsilon = \rho^4$ if we had perfect approximation of the corresponding generators.

## F  ETTH1: ADDITIONAL EXPERIMENT

The ETTh1 dataset Zhou et al. (2021a) is originally intended for forecasting a single multivariate time series measured on a regular time grid and therefore provides no independent trajectory samples. To adapt it to a generative modeling setting, we segment the long sequence into shorter overlapping trajectory chunks of fixed length 200 (using at most 80% overlap). These chunks are then assigned to training (4800 trajectories), validation (600 trajecotries), and test (600 trajectories) sets in such a way that overlaps occur only within a split and never across different splits, preventing information leakage between nearly identical sequences. This modification is minimal but necessary to obtain samples for density estimation and generator matching. We randomly generated the subsampled trajectories similar to the other data sets with rates of 25, 50 and 100 percent, and train our models for $\rho^2 = 0.001$ and $\eta^2 = 1$. For higher subsampling rates the performance is slightly reduced, which we attribute to the need for additional hyperparameter tuning at these finer temporal resolutions. We report the results in Table 5 and illustrate the generated paths in Figure 4. For visualization, we display the models' runs that achieve the lowest Sinkhorn distance in validation among those with small MMD values. Since MMD is the primary evaluation metric used in the TFM benchmark, our quantitative comparison continues to rely on MMD scores.

| Method | 50 | 100 | 200 |
|---|---|---|---|
| Jump-based method | [**0.642** ± 0.102] | [**0.885** ± 0.127] | [**1.731** ± 0.447] |
| SDE-based method | [1.819 ± 0.502] | [3.142 ± 0.967] | [3.069 ± 0.562] |
| TFM method Zhang et al. (2024) | [1.090 ± 0.462] | [1.984 ± 0.485] | [1.786 ± 0.533] |

Table 5: Average MMD over training runs between generated and ground truth trajectories for different methods on the ETTh1 dataset. Memory length $m = 20$ is used.

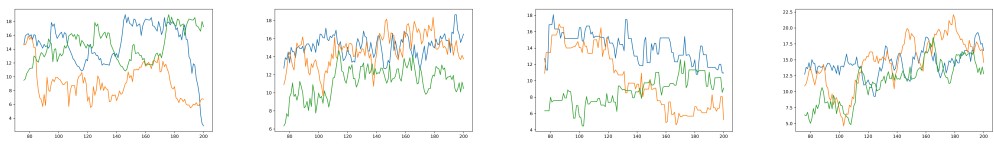

Figure 4: Results on the ETTh1 dataset with 25% subsampling rate. From Left to Right: Ground truth, TFM, JUMP, SDEM.

# G    ADDITIONAL NUMERICAL DETAILS

In this section we give additional material for the experiments described in the numerics section. We use the following abbreviations: TFM - trajectory flow matching method from Zhang et al. (2024), SDEM - our method SDE based method, JUMP - our method with jump kernel, MS-$\alpha$ - Markov superposition with parameter $\alpha$. In the following graphics we display the plots for the best Markov superposition $\alpha$, which was determined by a parameter search via a validation set.

**Implementation Details**    For all experiments we use a standard MLP architecture with ReLU activation functions, with 256 neurons and four hidden layers. We train using a learning rate of $1e - 5$ for 300 epochs. We evaluate the MMD between generated and ground truth samples (from our validation set) as a validation loss after every epoch and save the network with the best validation loss. For futher details see the code in the supplementary material.

**Synthetic dataset**    The synthetic data set is constructed as follows. Each trajectory consists of $T$ discrete timesteps $(x_i, t_i)$ for $i = 1, .., T$. We define a deterministic trend function over time, given by $(t/T)^2$ for $t = 0, 1, \ldots, T - 1$. Half of the trajectories follow a positive trend and the other half a negative trend, by multiplying the trend by $+1$ or $-1$, respectively. The initial values of each trajectory are sampled independently from a standard normal distribution. For each subsequent timestep, the value is updated by the following scheme: With probability $0.8$, the next value is computed as the sum of the current value and the trend at the next timestep. With probability $0.2$, the next value is the sum of the current value, the trend at the next time step, and a noise term drawn from $\mathcal{N}(0, 0.5^2)$. Here we take 50 time steps and use $\eta^2 = 0.3$ and $\rho^2 = 0.03$ throughout.

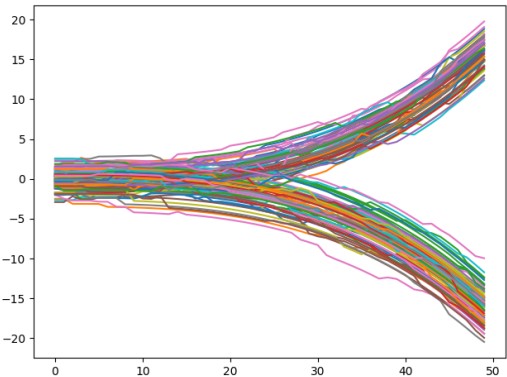

Figure 5: Ground truth data

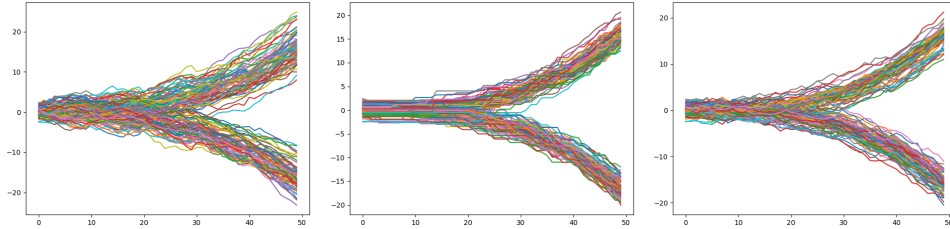

Figure 6: Results on the synthetic dataset using a $10\%$ subsampling rate. From Left to Right: TFM, JUMP, SDEM. The best $\alpha$ for the Markov superposition of the SDEM and the JUMP is zero, which is why it coincides with the SDEM

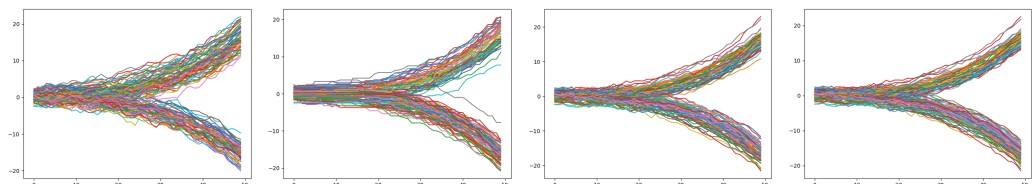

Figure 7: Results on the synthetic dataset using a 20% subsampling rate. From Left to Right: TFM, JUMP, SDEM, MS-0.85.

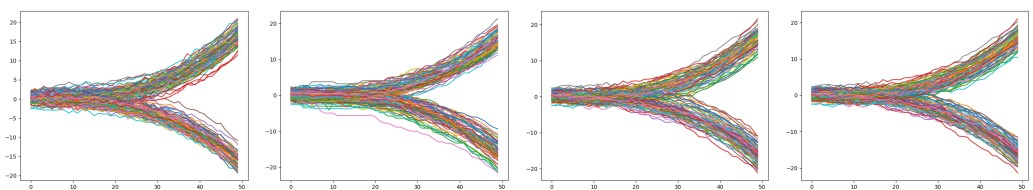

Figure 8: Results on the synthetic dataset using a 50% subsampling rate. From Left to Right: TFM, JUMP, SDEM, MS-0.90.

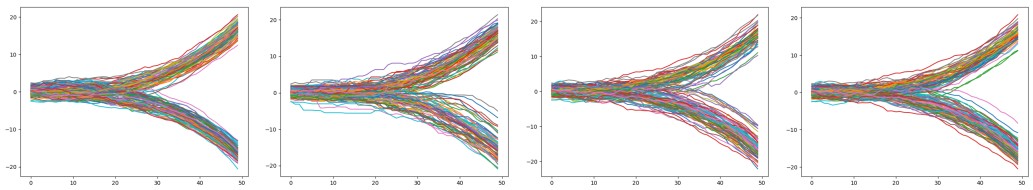

Figure 9: Results on the synthetic dataset using a 100% subsampling rate. From Left to Right: TFM, JUMP, SDEM, MS-0.80.

**Simulated Black-Scholes dataset** We created the data with the code from Herrera et al. (2021). Further implementational details can be found in the supplemental code, below we present additonal images.

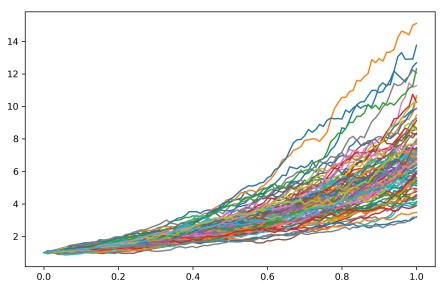

Figure 10: Ground truth data

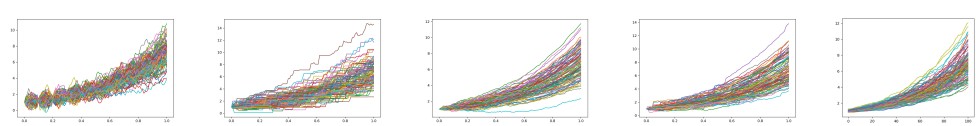

Figure 11: Results on the synthetic Black-Scholes dataset using a $10\%$ subsampling rate. From Left to Right: TFM, JUMP, SDEM, MS-$0.90$, SDEMatching.

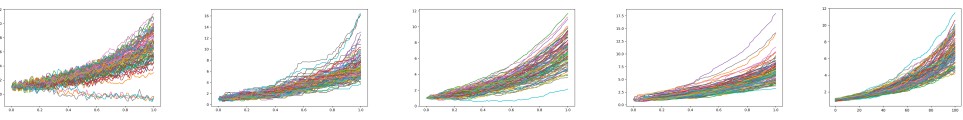

Figure 12: Results on the synthetic Black-Scholes dataset using a $25\%$ subsampling rate. From Left to Right: TFM, JUMP, SDEM, MS-$0.80$, SDEMatching.

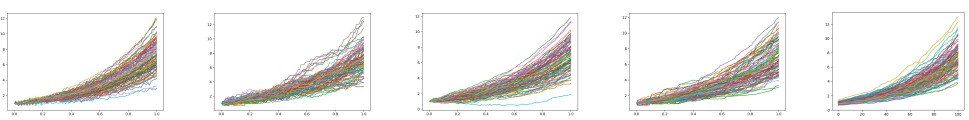

Figure 13: Results on the synthetic Black-Scholes dataset using a $50\%$ subsampling rate. From Left to Right: TFM, JUMP, SDEM, MS-$0.7$, SDEMatching.

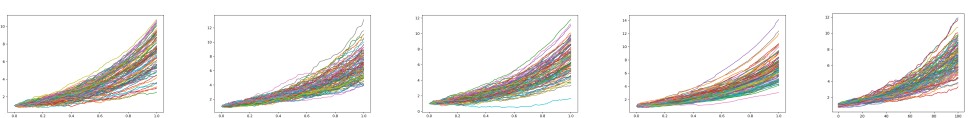

Figure 14: Results on the synthetic Black-Scholes dataset using a $100\%$ subsampling rate. From Left to Right: TFM, JUMP, SDEM, MS-$0.9$, SDEMatching.

**Moving Box Experiment.** For the moving box experiment we created the following dataset. We start with a 3x3 box with pixel value 1 anywhere in the 16x16 image which has pixel value 0 otherwise. We then draw a random direction, either left or right, and a random vertical direction either up or down. We then move the box one pixel in the horizontal direction and, with a probability of $30\%$, in the vertical direction. Every time the box would leave the image, we just reverse the respective direction and perform the update with respect to this new direction.

We consider a time series of 16×16 binary images, where each state is a vector $x_t \in \{0, 1\}^{256}$. Our algorithm is applied component-wise, see (Holderrieth et al., 2025, Section 7.2, Proposition 5 and Appendix F). When creating a new moving box time series, we update the memory by binarizing the output of the network by setting the highest 9 pixel values to $1$ and the rest to $0$. This ensures that the memory only includes binary images, which is important since the network was only feed with binary images during training. The binarized output does not always form a contiguous 3×3 box (see Figure 16 for an example). The learned dynamics are shown in Figure G. We quantify the frequency of this issue in Table 6. Importantly, Figure 16 also shows that such local errors do not necessarily destabilize the overall time series dynamics.

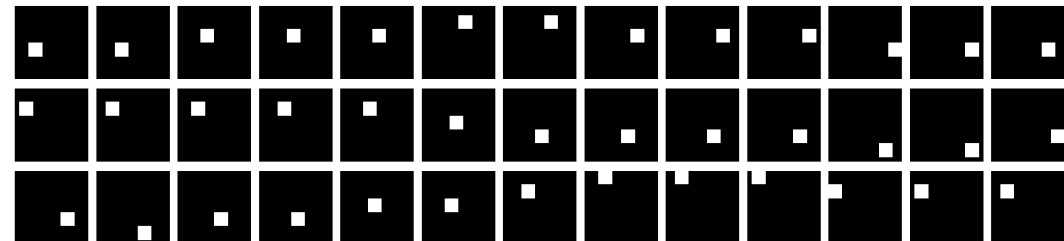

Figure 15: Learning of the dynamics of a moving box. Trajectories are given row-wise: one ground truth trajectory (top row), one trajectory generated via a learned drift-diffusion generator (middle row) and via a learned jump generator (bottom row).

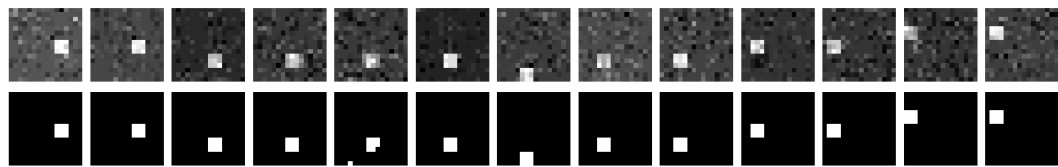

Figure 16: Time series of moving boxes. Top: output of the network. Bottom: binarized output.

Table 6: Full summary of autoregressive statistics, averaged over 5 seeds. Values are shown as Mean % ± Standard Deviation computed over 5 seeds.

| $\sigma$ | $\alpha$ | Well-Formed Cube | Correct X direction Movement | Y-Jump Ratio | Incorrect Y Jumps |
|---|---|---|---|---|---|
| | | | **Metrics** | | |
| | 0.0 | $86.74 \pm 1.60$ | $95.07 \pm 0.70$ | $31.24 \pm 3.45$ | $2.26 \pm 0.96$ |
| | 0.1 | $86.54 \pm 1.34$ | $95.65 \pm 0.80$ | $33.00 \pm 2.55$ | $2.33 \pm 1.08$ |
| | 0.2 | $84.83 \pm 1.96$ | $95.73 \pm 0.81$ | $34.24 \pm 3.15$ | $2.65 \pm 0.90$ |
| | 0.3 | $82.24 \pm 1.84$ | $95.53 \pm 0.73$ | $34.33 \pm 3.39$ | $3.03 \pm 0.99$ |
| | 0.4 | $80.23 \pm 1.63$ | $95.72 \pm 0.71$ | $33.97 \pm 3.68$ | $3.49 \pm 1.43$ |
| 0.05 | 0.5 | $82.22 \pm 1.43$ | $96.79 \pm 0.71$ | $34.41 \pm 3.53$ | $3.55 \pm 1.81$ |
| | 0.6 | $87.48 \pm 0.62$ | $98.15 \pm 0.56$ | $35.17 \pm 4.83$ | $3.52 \pm 1.63$ |
| | 0.7 | $93.26 \pm 0.59$ | $98.94 \pm 0.46$ | $36.62 \pm 6.29$ | $3.10 \pm 2.18$ |
| | 0.8 | $97.35 \pm 0.39$ | $99.49 \pm 0.41$ | $36.49 \pm 7.07$ | $2.93 \pm 2.20$ |
| | 0.9 | $98.89 \pm 0.24$ | $99.60 \pm 0.35$ | $36.25 \pm 7.31$ | $2.99 \pm 2.20$ |
| | 1.0 | $99.66 \pm 0.19$ | $99.69 \pm 0.30$ | $36.07 \pm 8.77$ | $3.11 \pm 2.49$ |
| | TFM | $100.00 \pm 0.00$ | $100.00 \pm 0.00$ | $14.75 \pm 4.60$ | $1.05 \pm 0.48$ |
| | 0.0 | $90.70 \pm 4.67$ | $95.53 \pm 2.43$ | $29.30 \pm 4.13$ | $2.41 \pm 0.50$ |
| | 0.1 | $91.36 \pm 4.52$ | $96.43 \pm 2.01$ | $30.02 \pm 3.54$ | $2.72 \pm 1.12$ |
| | 0.2 | $90.88 \pm 4.58$ | $96.43 \pm 1.81$ | $30.17 \pm 2.88$ | $2.53 \pm 0.78$ |
| | 0.3 | $90.00 \pm 4.96$ | $96.51 \pm 1.69$ | $30.25 \pm 2.45$ | $3.13 \pm 0.62$ |
| | 0.4 | $89.27 \pm 5.36$ | $96.55 \pm 1.82$ | $30.50 \pm 2.11$ | $3.18 \pm 0.82$ |
| 0.1 | 0.5 | $89.14 \pm 5.53$ | $97.02 \pm 1.54$ | $30.30 \pm 2.27$ | $3.35 \pm 1.32$ |
| | 0.6 | $89.82 \pm 5.14$ | $97.33 \pm 1.37$ | $30.47 \pm 2.16$ | $4.38 \pm 1.18$ |
| | 0.7 | $91.85 \pm 3.92$ | $97.94 \pm 1.05$ | $31.38 \pm 2.00$ | $4.48 \pm 2.02$ |
| | 0.8 | $93.65 \pm 2.92$ | $98.40 \pm 0.90$ | $31.96 \pm 1.87$ | $4.94 \pm 1.94$ |
| | 0.9 | $95.31 \pm 1.85$ | $98.67 \pm 0.73$ | $32.10 \pm 2.87$ | $5.31 \pm 2.69$ |
| | 1.0 | $96.35 \pm 0.83$ | $98.94 \pm 0.61$ | $32.67 \pm 3.42$ | $5.58 \pm 2.49$ |
| | TFM | $100.00 \pm 0.01$ | $99.98 \pm 0.01$ | $38.21 \pm 2.85$ | $2.24 \pm 0.64$ |

