# OpenReview forum: "Learning Jump-Diffusion Dynamics from Irregularly-Sampled Data"
_ICLR.cc/2026/Conference — Submitted to ICLR 2026_

### Official Review · Reviewer_7wcJ · 2025-10-20

**Soundness:** 3
**Presentation:** 1
**Contribution:** 3
**Rating:** 2
**Confidence:** 2

**Summary:**

A paper proposing a diffusion model for irregularly sampled time series. The key innovation is the proposal of jump-diffusion transition kernels to model the bridge probability in a way that is independent of the sampling time of the data. The paper is primarily theoretical and has a limited evaluation on synthetic data, but it does use one semi-realistic data set consisting of a moving square.

**Strengths:**

A mathematically rigorous paper trying to extend generative diffusion modelling to a non-trivial scenario of irregularly sampled time series. The quality of the work is sound and the mathematical details are given in depth.

**Weaknesses:**

- The paper is way too dense to expect someone who is not in the same niche to be able to grasp much, particularly given the very short time given to reviewers (who also have another four papers and, occasionally, a day job). I appreciate that the topic is highly technical, but the notation and style of presentation would have been significantly improved by some streamlining.
- The main technical contribution appears to be a regularisation in the bridge probability, is that so? It is somewhat difficult to assess exactly what was already done in the very recent papers by Holderrieth and Zheng, and what was innovation in this paper.
- Sometimes there is some sloppiness in the use of the word "process", for example Alg. 1 is a procedure to sample trajectories (approximately), which in most literature is distinct from the process (infinite dimensional object).
- The memory aspect is another place where greater clarity would be needed, it's even unclear what it's meant (my impression was that the statistics of the project were obtained by some moving average over the trajectory but  as the topic is important its relegation to a short paragraph was definitely too little)
- The empirical section was very schematic, which might be fine for a strongly theoretical paper but perhaps something that would showcase better the advantages of including jumps would have helped, as the performance is often close to the diffusion method.

**Questions:**

- It seems to me that the trajectories might be irregularly sampled, but all trajectories must be sampled at the same times, is that so?
- Could you clarify a bit more how memory is introduced?

---

> ### Author Response · Authors · 2025-11-20
> **Answer to Reviewer 7wcJ**
>
> Thank you very much for your thoughtful and detailed review.
> In addition to the general response above, please find below our point-by-point replies to your comments and questions.
>
> Regarding weakness 1, to ensure the rigor and correctness of our theoretical results, it was unfortunately unavoidable to rely on precise mathematical formulations.
>
> Regarding weakness 2, the regularization of reference bridges is only one aspect of our contribution beyond Holderrieth and Zhang. In addition, we introduce parametric jump measures that allow closed-form computation of the training losses and derive theoretical guarantees establishing both consistency and robustness of the proposed estimator. Further details are provided in the general response above.
>
> Regarding weakness 3, Algorithm 2 describes approximate sampling from the joint distribution of $(X_t)_{t\in[0,1]}$. The number of sampling points $n$ and the corresponding times $t_1,\ldots,t_n$ are arbitrary and together determine the distribution of the process. Thus, while the algorithm operates on a finite discretization, it remains consistent with the underlying stochastic process in law.
>
> Regarding weakness 4, the “memory’’ mechanism refers to conditioning on past trajectory states to capture dependencies that go beyond strict Markovianity. This provides a more flexible modeling framework for systems whose future evolution depends on longer temporal context, such as mixtures of Markov processes or processes with temporal correlations. Since pure Markovianity is often too restrictive for real data, the inclusion of memory serves as a principled relaxation. We have expanded the explanation of this concept in the revised manuscript.
>
> Regarding weakness 5, empirically the combination of jump and diffusion (SDE) components generally performs best, often by a noticeable margin. These results suggest that incorporating jumps can be advantageous even in Euclidean spaces, which remain underexplored in generative modeling. Given that such jump-based generators have only recently become accessible, we believe their performance justifies further research in this direction.
>
> Below are the answers to your questions:
>
> - **On the irregularity and variation of the observation times:** The trajectories are irregularly sampled, and two trajectories from the same dataset typically have different observation times (while maintaining the same total number of observations). Handling both irregular and non-aligned sampling directly is challenging, as most existing methods rely on preprocessing steps to homogenize the data. Our framework can address this case directly without such preprocessing.
>
> - **On the memory**: We have extended the discussion of the memory mechanism in the revised paper. In short, the memory captures dependencies on past states to enhance representational flexibility and to model non-Markovian dynamics more accurately.

---

> > ### Comment · Reviewer_7wcJ · 2025-11-26
> >
> > Thanks for your detailed response, I'll read the revised version. My comment on weakness 1 was not to detract from mathematical rigour, which I agree is necessary; nevertheless I also think an effort should be made to ensure consistency and ease of notation, and choices need to be made as to which material needs to be presented in the main manuscript, and which can be put in an appendix, so that the logical flow of the work is easily accessible.

---

> > > ### Author Response · Authors · 2025-11-28
> > > **Answer to comment by Reviewer 7wcJ**
> > >
> > > Thank you for this constructive comment. We fully agree that clarity, consistency, and ease of notation are important. In the revised version, we have therefore added a concise high-level paragraph Appendix B that summarizes our approach and its key ideas and created two new Figures 1 and 2 visualizing our derived interpolating jump and diffusion bridges, while keeping the main text focused on the logical structure of the method. We are also happy to further streamline the exposition and would greatly appreciate any concrete suggestions for places where notation or presentation could be simplified.
> > >
> > > In addition, following the suggestion of Reviewer VY4R, we have included experiments on the real-world ETTh1 dataset to complement our synthetic benchmarks, see Figure 5 and Table 4 in the new section F. We believe this further improves the accessibility of the work and demonstrates the applicability of our method beyond controlled settings.
> > >
> > > We hope these changes address the concern and improve the overall flow of the manuscript.

---

### Official Review · Reviewer_VY4r · 2025-10-27

**Soundness:** 2
**Presentation:** 3
**Contribution:** 2
**Rating:** 4
**Confidence:** 4

**Summary:**

The authors apply a generator learning technique to the task of irregular time series generation. The authors derive the neccessary loss for such an application, and show performance in synthetic settings.

**Strengths:**

1. The work is technically solid, the derivations are well written and clearly presented.
2. The writing and presentation of the paper is clear, it is easy for readers to understand.

**Weaknesses:**

Minor:
1. Slight notation inconsistency, e.g $\Delta$ and $\nabla^2$ are both used for hessian in different places.
2. The paper assumes an identity diffusion coefficient, limiting the modeling flexibility. Although this is relatively standard in this type of research, the diffusion coefficient can be easily modeled by a neural network.

Major:
1. The work lacks novelty. This is perhaps the most concerning weakness. The work seems to be taking Holderrieth 2025's idea and apply it to the irregular time series modeling setting in TFM (Zhang, 2024).
2. Related to the previous points, it will be more convincing that the work is empirically solid by repeating some numerical experiment in TFM. Most of the experiments now only show that it improves in synthetic scenarios.
3. The first two points combined lead to a relatively weak motivation to apply generator learning (with jump processes) in such a specific field of application.

**Questions:**

1. Can the author explain why having no singularities during the observed points of irregularly sampled data is beneficial? Is it solely to construct a full probabilistic view, or are there some particular benefits to having $P_0, P_1$ not being the observed points, i.e does it improve robustness, help with uncertainty quantification, etc?
2. Can the author explain why having a jump process during the generator learning is necessary? Aside from the benefit during synthetic data modeling, does it actually help (or perhaps at least does not deteriorate) in modeling full continuous time series?
3. The parameter $\eta^2 = 0.3$ seems small compared to the synthetic data's scale, leaving a rather smooth path to model; doesn't this limit the usage of SDE? What happens if we leave $\eta^2 = 1$?

---

> ### Author Response · Authors · 2025-11-20
> **Answer to Reviewer VY4r**
>
> Thank you very much for your careful and constructive review.
>
> **Regarding the minor weaknesses:**
>
> We have corrected the inconsistency in the notation for the Hessian, thank you for pointing this out.
> We agree that assuming an identity diffusion coefficient restricts modeling flexibility. However, our paper already extends one-dimensional paths to higher dimensions through factorized trajectories, which effectively allows for distinct coefficients across dimensions. Extending the model further by learning the diffusion coefficient is indeed a natural next step. We note, however, that this extension is nontrivial: since the data are irregularly sampled with variable observation times, one must derive a suitable supervised training signal from such data, which renders many standard estimation strategies infeasible in our setting.
>
> **Regarding the major weaknesses:**
>
> Please refer to the general response above for a detailed discussion of the substantive extensions we make beyond the works of Holderrieth and Zhang. We emphasize that jump models in Euclidean spaces remain largely unexplored, and thus improvements even in synthetic scenarios are scientifically valuable, in particular as several issues of the first numerical implementations in Holderrieth are resolved.
>
> We would have welcomed reproducing experiments from TFM, however, the corresponding medical dataset and most of the TFM source code are not publicly available. The Jupyter notebook provided by the TFM authors serves only for qualitative illustration and uses entirely synthetic, deterministic harmonic-oscillator trajectories sampled on a uniform grid. Consequently, a direct numerical comparison was not feasible within this study.
>
> **Answers to specific questions:**
>
>  - **On the absence of singularities:**
>     Avoiding singularities in generative models is crucial for numerical stability and robust training. In practice, many methods handle this implicitly, for example, by sampling $t \in [0, 1 - \varepsilon]$ instead of $[0,1]$, interpolating via $(1 - (1 - \varepsilon)t)x_{latent} + t x_{data}$ in flow matching, or truncating diffusion processes at finite times even though the theoretical limit requires $t \to \infty$. Removing singularities improves robustness and prevents divergence in gradients or likelihoods. Interpreted probabilistically, it can also be viewed as introducing an augmented prior that may benefit uncertainty quantification and would be interesting to explore further.
>
> - **On the role of the jump process:**
>     Incorporating a jump process provides additional flexibility and is more appropriate when the underlying continuous-time dynamics exhibit discontinuities. Jump processes in Euclidean space remain comparatively underexplored within generative modeling, despite being a natural complement to the well-studied SDE-based methods. Including jumps therefore broadens the representational capacity of generator learning without deteriorating performance on smooth data, as the model can revert to the diffusion-dominated regime when jumps are unnecessary. Note that also if the underlying data has continouos trajectories, using a pure jump process for generation does not deteriorate, but might be slightly worse compared to a pure sde model, see table 2 and 3.
>
> - **On the choice of $\eta^2$:**
>     The case $\eta^2=1$ was included in our hyperparameter search over $\{10,3,1,0.3,0.1,0.03,0.01, 0.003\}$, see the supplementary code. The corresponding validation errors for $\eta^2=1$ are $0.224,0.232,0.233$ and $0.336$ instead of $0.051, 0.054, 0.065$ and $0.066$ (for $\eta^2=0.3$) for the different subsampling rates.

---

> > ### Comment · Reviewer_VY4r · 2025-11-26
> >
> > I understand now that extending the framework of generator matching to irregularly sampled time series with jumps is non-trivial and technically novel. However, the empirical experiments still lack realistic applications. The author should strengthen the experiments to fully convince me. One way to benchmark results is to take some regularly sampled time series benchmarks (etth1 or some common dataset), sample them irregularly, and then demonstrate goodness of fit. It would be possible to compare this result to some other time series methods. This will at least allow the reviewers to evaluate how the method can work beyond synthetic scenarios.

---

> > > ### Author Response · Authors · 2025-11-28
> > > **Answer to Comment by Reviewer VY4r**
> > >
> > > Thank you for your feedback and that you appreciate the novelity and difficulty of our approach. Following your recommendation, we have added experiments on the ETTh1 benchmark as a non-synthetic example. Because ETTh1 is originally designed for forecasting a single long time series, we applied a minimal adaptation to make it suitable for generative modeling: we segmented the series into trajectory samples and constructed train/val/test splits that avoid overlap across groups. This allows us to use ETTh1 as a proper sample-based generative modeling benchmark, see newly added section F with Figure 4 and Table 5.
> > >
> > > Using this adapted dataset, for consistency we run the same generative modeling experiments as for the toy and Black–Scholes settings. The results, see  Table 5, show that the jump model can is beneficial for this real dataset. Full details of the dataset adaptation are provided in the new Section F.
> > >
> > > We also note here that following Reviewer 7wcJ suggestions' we have added new Figures 1 and 2 for improving clarity and illustrating our approach, and a high-level roadmap in Appendix B. In particular, the new Figure 1 (left) visualizes how our regularization of paths regularizes endpoint singularities.

---

### Official Review · Reviewer_gjfR · 2025-10-30

**Soundness:** 3
**Presentation:** 2
**Contribution:** 2
**Rating:** 4
**Confidence:** 2

**Summary:**

The authors propose a method that estimates jump diffusions from observational data. By using existing known generators of certain classes of jump diffusion processes, the authors propose transforming those such that their time marginals match the observational data. This is similar to the techniques used in stochastic interpolants which maps one marginal distribution with known properties to another. The authors leverage an Ito-Levy type decomposition to represent the continuous and pure jump parts. The key component is to parameterize the jump component as a Gaussian distribution which allows the use of KL divergence for training. The authors then describe how to estimate the related generator given observations of irregularly sampled time series. Finally, the authors consider a few numerical examples on the proposed method.

**Strengths:**

The authors propose an elegant formulation that matches a given base generator to a target generator that supports jump processes.

The authors find initial distributions that are amenable to sampling from.

The authors find a workaround to compute the jump measure more effectively than in existing works.

**Weaknesses:**

The central contribution is largely a combination of the work of Zhang et al and Holderrieth et al to consider generator matching with jumps.

The numerical results appear to be inconclusive as to the efficacy of the method, which is not something to be concerned about, but it would be good to see which scenarios the method does perform well in as a comparison. This leads to a question on where the method should be employed what set of tasks. This was not evident in the main text.

The numerical experiments do not illustrate the importance of the jump component, which should be studied.

**Questions:**

Can the authors detail the differences and the techniques needed to bridge Zhang el al to Holderrieth et al?

I’m struggling a bit to understand what the correct use-case of this method would be. The authors motivate with financial data or possibly limit order book data which is irregularly sampled, but there are a series of methods that could work in such scenarios. Can the authors comment on this a bit more, why and where the particular method would work well?

Are there more obvious jump related data that the authors could consider? This would go a long way in motivating the use case of the method.

Is it limiting to consider functions $r$ that are parameterized by a Gaussian?

---

> ### Author Response · Authors · 2025-11-20
> **Answer to Reviewer Reviewer gjfR**
>
> Thank you very much for your detailed review.
>
> Regarding weaknesses (1) and (2), please refer to the general response above, which discusses in detail the paper’s central contributions, the technical distinctions between Zhang et al. and Holderrieth et al., and the practical scope of our approach.
>
> Concerning weakness (3), we would like to highlight Table 3 added to the main body, which reports results on the  two-dimensional BS dataset. There, even the pure jump model (with factorized components) alone outperforms the diffusion-only (SDE) model. Similarly, in Tables 1 and 2, the best-performing configuration for irregularly sampled data is consistently the Markov superposition model combining jump and diffusion components, although the margin is smaller. Note also that the learned weighting factor $\alpha$ assumes values between $0$ and $0.9$ for the different experiments. Note that heuristically, $\alpha \le 0.5$ indicates that the majority of the overall performance improvement can be attributed to the jump component.
>
> Below we provide specific answers to your four questions:
>
>
> - **On bridging Zhang et al. and Holderrieth et al.:**
> As detailed in the general response, our work (i) replaces the binned estimation of jump measures by a parametric family that admits closed-form training losses, (ii) introduces regularized reference generators to avoid endpoint singularities, and (iii) establishes a consistency result showing that training on subsampled trajectories recovers the correct joint path distribution.
>
> - **On use-cases and jump-related data:**
> As discussed above, the proposed framework is particularly effective for irregularly sampled systems that exhibit discontinuities, settings where continuous diffusions systematically underestimate tail risks. Relevant examples include financial series with jumps, neuronal spiking activity, and abrupt regime changes in physical systems. Our method uniquely combines irregular sampling with non-continuous latent dynamics in a single generative formulation, a feature that competing approaches generally lack. We agree that further applications to clearly discontinuous empirical datasets would strengthen motivation; however, we focused here on synthetic and semi-realistic cases to isolate theoretical aspects and verify correctness.
>
> - **On Gaussian parameterization:**
> Using a Gaussian parameterization is not a fundamental limitation but a tractable and analytically convenient starting point. It allows closed-form KL losses and ensures stable optimization. The formulation extends naturally to richer parametric families, particularly exponential families with known sufficient statistics, since the KL divergence then remains expressible through their parameters. Extensions beyond exponential families also appear feasible within our framework.

---

> > ### Author Response · Authors · 2025-11-28
> > **Comment to Reviewer gjfR**
> >
> > We wanted to kindly follow up, as we have not yet received feedback on our initial response which adressed all your questions and concerns in detail.
> >
> >  Moreover, in the meantime, we have carried out substantial updates to the manuscript based on the detailed comments of the other reviewers. These include new illustrative figures describing the method, an expanded high-level overview of the method, clarifications of definitions, and additional experiments (including a real-data evaluation on ETTh1).
> >
> > We would greatly appreciate any further feedback you might have.

---

### Author Response · Authors · 2025-11-20
**General Answer**

We thank the reviewers for their constructive and helpful comments.
We have revised the manuscript carefully to address all concerns and looking forward for discussions.
Modifications in the revised version are marked in blue. Apart from this general answer please find individual answers to each report below.

Sorry for not clearly committing the new contributions of our paper. We like to emphasize that our work **goes far beyond a simple combination of Generator Matching and Trajectory Flow Matching (TFM)** which we explain briefly below.
We have rewritten the introduction accordingly.


- If  generator matching is straightforwardly applied
to irregularly sampled time series, **a functional version of generator matching would be required**. To the best of our knowledge, such a version has only been explored in the context of functional matching for deterministic flows (cf. Kerrigan et al.), and on equispaced data. In contrast, we adopt an autoregressive perspective  and **approximate the generator locally in time**. This introduces several nontrivial challenges that are specific to jump processes. Unlike flow-based models, where the objects of interest are vectors, **jump processes involve measures, which require an entirely different treatment of the loss function and optimization**.

- **In standard generator matching, jump measures were approximated by naive binning**, which scales poorly and prevents closed-form computation of the training losses. These losses must then be evaluated numerically, which is inefficient and unstable.
**Our contribution is to approximate the jump measures by a parametric family of distributions for which the losses can be derived analytically (see Propositions 5 and 6)**. This provides a substantial computational speed-up and a more stable training objective.

- A second fundamental drawback of both generator matching and TFM is that the reference generators for both the diffusion and jump components **become singular at the endpoint of the process**. This singularity arises because Gaussian base distributions have unbounded support, while empirical data typically lie on compact domains.
**We address this by modifying the reference processes: specifically, we augment the data with small perturbations (Proposition 2)** and incorporate it formally into the theoretical derivation of the reference processes and to provide exact modified formulations.

- We offer **the first mathematically precise explanation of why the memory mechanism works (Proposition 7)**  and  give theoretical justification for the use of memory in irregularly sampled generator training.

Secondly we comment here on potential **use-cases of the method**, which is a second issue raised by the reviewers. We emphasize that **jump processes in Euclidean space represent an entirely new class of generative models**. So far this was only considered in Holderrieth et al., and their exploration is still at an early stage.  Our work introduces several theoretical and computational improvements that make these models practically viable.

 **The broader motivation for incorporating jumps into time series modeling is that many real-world stochastic systems exhibit genuine temporal discontinuities.** Modeling these with purely continuous diffusions introduces bias and underestimates tail risk. Examples could be found in finance (sudden price jumps, volatility spikes), neuroscience (spiking neural activity) or climate/physical systems (abrupt regime shifts or threshold-driven events). Our method lays the basis for practical applications where both irregular sampling and discontinuous latent dynamics are intrinsic features of the data. **Empirically the combination of jump and diffusion (SDE) components generally performs best in our examples, often by a noticeable margin.**

---

### Meta-Review · Area_Chair_aNtN · 2025-12-15

**Summary:**

The reviewers agree the submission is technically serious and mathematically detailed, proposing to extend generator matching to irregular, non-aligned time series and to incorporate jump components in addition to diffusion. The paper derives explicit stabilized interpolation bridges (diffusion and jump) and trains conditional generators using a memory window; it also proposes a Gaussian-parameterized jump kernel to obtain closed-form KL losses in low dimensions and motivates superposition of jump+diffusion generators.

However, the decision is driven by several consistent concerns:
	•	Insufficient empirical validation / unclear practical impact: The original evaluation was viewed as largely synthetic and not clearly demonstrating when jumps are necessary or beneficial relative to simpler diffusion-only or existing baselines. While the paper includes multiple synthetic studies and a moving-box dataset, the extent to which this supports broad claims about real irregular time series remains limited (and some reported comparisons still show strong performance of the baseline TFM in parts of the evaluation).
	•	Novelty concerns: Multiple reviewers felt the core idea appears close to combining recent “trajectory flow matching / stochastic interpolants” style conditional bridges with generator matching with jumps, and it was initially difficult to disentangle what is fundamentally new vs. an application/extension of very recent works.
	•	Presentation density and clarity: Reviewer 7wcJ in particular flagged that the manuscript is too dense, with heavy notation and limited high-level guidance, making it difficult to evaluate and lowering accessibility. The paper itself acknowledges this and points readers to a high-level roadmap appendix.

Overall, the paper has promising technical elements, but the evidence and positioning are not yet convincing enough for ICLR acceptance, primarily due to limited compelling real-world demonstrations and remaining ambiguity about incremental novelty vs. contemporaneous work.

**Reviewer Concerns:**

Concerns substantially addressed by rebuttal / revision
	•	Use-case motivation and “why jumps”: The rebuttal gave a clearer rationale for jump processes (discontinuities, tail risk, irregular sampling) and framed plausible domains (finance, spiking activity, regime shifts). This improves the story, though it does not fully substitute for strong empirical validation on such datasets/domains.
	•	Clarifying technical distinctions: The authors clarified claimed contributions beyond a straightforward combination: (i) stabilized reference bridges to avoid endpoint singularities by adding small perturbations; (ii) parametric jump-measure approximation enabling analytical training losses; (iii) theoretical discussion/consistency around memory-based conditioning and joint-path recovery. These map to concrete components in the paper (stabilized bridges in Prop. 2 / Remark 3; Gaussian KL for jumps in Prop. 5–6; memory conditioning and Prop. 7).
	•	Added real-data experiment: In response to discussion, the revision adds an ETTh1 experiment in the appendix (segmented into trajectory chunks) and reports MMD comparisons where the jump-based method can outperform TFM/SDE variants in that setup. This is a positive step.


Concerns still outstanding
	•	Empirical strength remains below ICLR bar: Even with ETTh1 added, the real-data evaluation is (a) only one dataset, (b) relies on a specific adaptation (chunking a single long series), and (c) uses primarily MMD-based evaluation; it does not convincingly establish broad usefulness or superiority, nor does it include strong, diverse baselines standard in irregular time-series modeling (e.g., latent ODE/SDE variants, neural CDE, modern diffusion/flow time-series generators) beyond limited comparisons and discussion. The experiments section itself notes other baselines but does not include them.
	•	Demonstrating necessity of the jump component: The paper argues jump+diffusion can help and provides cases (e.g., synthetic BS 2D and moving-box vertical jumps). But the evidence remains mixed: for the moving-box dataset, TFM is extremely strong on several metrics, and the “jump advantage” is mainly visible on specific discontinuity-related statistics; this does not fully establish a compelling practical win for the more complex machinery.
	•	Scope/limitations of the jump-kernel approach: The closed-form jump KL and moment computations are restricted to low dimensions (explicitly d∈{1,2}) and a Gaussian(-like) family, which raises questions about scalability/general applicability for multivariate real time series; the paper positions higher-dimensional handling via factorization/component-wise application, but this may be limiting.
	•	Accessibility remains a concern: While the revision adds a roadmap appendix and figures, the core manuscript remains mathematically dense; reviewers indicated this impacts evaluability and perceived readiness.

Given these remaining issues, the rebuttal improves clarity and adds incremental evidence, but not enough to overturn the overall low evaluation.

**Reviewer Scores:**

Here is how I expect scores would move (or not) with full discussion:
	•	Reviewer gjfR (4: marginally below threshold; low confidence)
Likely stays at 4. The rebuttal improves articulation of contributions and adds ETTh1 + stronger claims about jump utility, which might slightly increase comfort, but novelty concerns and inconclusive practical positioning likely prevent a clear upgrade.
	•	Reviewer VY4r (4: marginally below threshold; confidence 4)
The reviewer explicitly acknowledged the technical novelty in discussion but maintained that realistic applications are still lacking, and requested benchmarking; after the authors added ETTh1, they might move from 4 → 5 (weak accept) at most if they found the new section convincing. However, given the limited breadth (single dataset, limited baselines), I think most likely remains at 4, possibly “4 but leaning up.”
	•	Reviewer 7wcJ (2: reject; low confidence / first-time reviewer)
The rebuttal addresses clarity concerns somewhat (new figures/roadmap; expanded explanation of memory), but the reviewer’s core objections included density and insufficient empirical showcase of jumps. I expect 2 → 3 (still reject) if they appreciate the added guidance and ETTh1, but unlikely to reach borderline accept without broader empirical support.

---

### Decision · Program_Chairs · 2026-01-26

Reject